

# A parallel workflow implementation for PEST version 13.6 in high-performance computing for WRF-Hydro version 5.0: a case study over the Midwestern United States

Jiali Wang, Cheng Wang, Andrew Orr, Rao Kotamarthi

Argonne National Laboratory, Environmental Science Division, 9700 South Cass Avenue, Argonne, IL 60439, USA

*Correspondence to*: Jiali Wang (jialiwang@anl.gov)

**Abstract.** Surface hydrological models must be calibrated for each application region. The Weather Research and Forecasting Hydrological system (WRF-Hydro) is a state-of-the-art numerical model that models the entire hydrological cycle based on physical principles. However, as with other hydrological models, WRF-Hydro parameterizes many physical processes. As a result, WRF-Hydro needs to be calibrated to optimize its output with respect to observations. However, when applied to a relatively large domain, both WRF-Hydro simulations and calibrations require intensive computing resources and are best performed in parallel. Typically, each physics parameterization requires a calibration process that works specifically with that model, and is not transferrable to a different process or model. Parameter Estimate Tool (PEST) is a flexible and generic calibration tool that can calibrate any numerical code. However, PEST in its current configuration is not designed to work on the current generation of massively parallel high-performance computing (HPC) clusters. This study ported the parallel PEST to HPCs and adapted it to work with the WRF-Hydro. The porting involved writing scripts to modify the workflow for different workload managers and job schedulers, as well as developing code to connect Parallel-PEST to WRF-Hydro. We developed a case study using a flood in the Midwestern United States in 2013 to test the operational feasibility of the HPC-enabled parallel PEST. We then evaluate the WRF-Hydro performance in water volume and timing of the flood event. We also assess the spatial transferability of the calibrated parameters for the study area. We finally discuss the scale-up capability of the HPC-enabled parallel PEST to provide insight for PEST's application to other hydrological models and earth system models on current and emerging HPC platforms. We find that, for this particular study, the HPC-enabled PEST calibration tool can speed up WRF-Hydro calibration by a factor of 30 compared to commonly-used sequential calibration approaches.



## 1 Introduction

Hydrological models are important tools for research relevant but not limited to, water resource management, flood control, and hydrological response to climate change (Zanon et al., 2010; Papathanasiou et al., 2015). Conceptual hydrological models express hydrological processes in the form of abstract models that come from physical phenomenon and experience. Physically based hydrological models contain definite physical mechanisms to model the hydrological cycle, but many complex physical processes in these models are parameterized. For example, the state-of-the-art Weather Research and Forecasting Hydrological modeling system (WRF-Hydro; Gochis et al., 2015) has dozens of parameters that can be land- and river-type dependent and are typically specified in lookup tables. Both conceptual hydrological models and physically based hydrological models need to be calibrated before they can be applied to research. In this context, calibration refers to the hydrologists' need to adjust the values of the model parameters so that the model can closely match the behavior of the real system it represents. In some cases, the appropriate value for a model parameter can be determined through direct measurements conducted on the real system. However, in many situations the model parameters are conceptual representations of abstract watershed characteristics and must be determined through calibration. In fact, model calibration is the most time-consuming step, not only for hydrological models but also for earth system model development, because both parametric estimation and parametric uncertainty analysis require hundreds—if not thousands—of model simulations to understand how perturbations in model parameters affect simulations of dominant physical processes and to find the optimum value of a single parameter.

WRF-Hydro is a practical physics-based numerical model that can simulate the entire hydrological cycle using advanced high-resolution data such as satellite and radar products. Compared to the traditional land surface model (LSM) used by WRF, WRF-Hydro provides a framework for multiscale representation of surface flow, subsurface flow, channel routing, baseflow, as well as a simple lake/reservoir routing scheme. As a physics-based model, WRF-Hydro includes many complicated physical processes that are nonlinear and must be parameterized. For example, the parameters for channel routing are prescribed as functions of stream order, not space; thus the default parameters given by WRF-Hydro are only valid over a small region. Because channel routing can affect the accuracy of the model performance, calibration of related model parameters



is often required in order to use the model in a new domain. In particular, for a large spatial domain such as the entire Contiguous United States (CONUS), in order to develop the optimal parameter sets in a reasonable amount of time, the calibration must be conducted on HPCs in parallel instead of in the traditional sequential mode. To date, there is no such calibration tool that can straightforwardly calibrate WRF-Hydro on HPCs. Typically, each physics-based model needs a calibration code that is custom-designed to work with that particular numerical model. These custom-designed calibration codes/tools are highly challenging and do not offer flexibility; they are designed to operate with that particular numerical model and its set of physics parameterizations, software architecture, and solvers. Therefore, there is a need for a more flexible and generic calibration tool that can calibrate any code that uses Message Passing Interface/Open Multi Processing (MPI/OpenMP) for parallelization on HPCs.

There are two general types of calibration methods for hydrological models: manual calibration and automatic calibration. Models for individual catchments have traditionally been calibrated by manually adjusting key model parameters within established ranges of parameters to obtain a best match between observed and simulated discharges. This procedure is time consuming, dependent on the skill and experience of the modeler, and therefore prone to inconsistency between modelers. Automatic calibration is based on stochastic or mathematical methods and thus is more widely applicable for optimizing nonlinear parameters. Compared with manual calibration, automatic calibration is more efficient and effective, because it avoids interference from human factors (Madsen, 2000; Getirana, 2010). One widely used automatic calibration tool is Parameter Estimation Tool (PEST; Doherty 2016), which uniquely operates independent of models. There is no need to develop additional programs/codes for a particular model except preparing the files required by PEST as described in Sect. 3.2, because PEST works with that model through the model's own input and output files. PEST implements a particularly robust variant of the Gauss-Marquardt-Levenberg method (Levenberg, 1944; Marquardt, 1963) to estimate parameters. This method requires a continuous relationship to exist between model parameters and model outputs, but it can normally find the minimum in the objective function in a fairly shorter time period than other parameter estimation methods. This is especially important when model runs are lengthy or when many parameters must be calibrated. Parallel PEST is able to distribute many runs across many computing nodes using master-slave parallel programing. However, to the best of our





knowledge, no approach is available that allows users to submit jobs using PEST parallelization
to a typical supercomputing facility that uses job scheduling and workload management using
Simple Linux Utility for Resource Management (SLURM), Portable Batch System (PBS), and
Cobalt. A previous study (Senatore et al., 2015) used PEST to calibrate WRF-Hydro over the Crati
River Basin in Southern Italy. However, because the study area was relatively small, they were
able to conduct the calibration using PEST in sequential mode.
In this study, we ported parallel PEST to HPC clusters operated by the U.S. Department of Energy
(DOE) and adapted it to work with WRF-Hydro. Porting involved writing additional scripts to
modify the workflow for SLURM, Cobalt, and PBS and developing code to connect parallel PEST
to WRF-Hydro. In particular, we aim to (1) calibrate the parameters of WRF-Hydro to improve
model performance with realistic values maintaining their physical meanings; (2) speed up
calibration for this particular study case and provide the capability to WRF-Hydro users; and (3)
explore the scale-up capability of HPC-enabled parallel PEST linked to WRF-Hydro.

## 2 Model description

### 2.1 Study area

The case presented here is one of the worst floods experienced by greater Chicago area in the last
three decades, which occurred on April 18, 2013 (Campos and Wang, 2015). According to the
National Weather Service (NWS), the heaviest 24-hour accumulated rainfall during this storm
reached 201.4, 171.1, and 136.4 mm across Illinois, Iowa, and Missouri, respectively. The
Mississippi River crested at 10.8 m (1.7 m above flood stage), and the Illinois River crested in
Peoria, Illinois, at 8.95 m; this broke the previous record of 8.78 m, set in 1943, and was 4.55 m
above the historical normal river stage (NWS, 2013). Campos and Wang (2015) conducted three-
domain nested WRF simulations to understand the dynamical and microphysical mechanisms of
the event. Our study builds on the smallest domain of that study, which covers the majority of
Illinois, Iowa, and Missouri at a spatial resolution of 3 km for the atmospheric and land surface
model (Fig. 1).



During the 10-day period of this studied case, light to moderate rain occurred on April 8 through
11, 2013, followed by a relatively dry period from April 12 to 14. Then a heavy rain event began
on April 15 and peaked on April 18. The heaviest rain band moved east of the study area on April
19. The rainy event ended over the study area on April 20.
**2.2 WRF-Hydro configuration**
WRF-Hydro employs a multiscale modeling approach to handle the local landscape gradient
features. Specifically, WRF-Hydro uses a subgrid disaggregation-aggregation procedure. For each
time-step at which forcing data are available, the column moisture stays within the LSM and is
disaggregated from the LSM grid to a high-resolution routing grid (Gochis and Chen 2003). After
disaggregation, the routing schemes are executed using the high-resolution grid values. After
execution of the routing schemes, the high-resolution grid values are aggregated back to the native
LSM grid. For details of each routing component, see Gochis et al. (2015), Yucel et al. (2015), and
Senatore et al. (2015).
Currently, two LSMs are available in WRF-Hydro for representing land-surface column physics:
Noah (Chen and Dudhia, 2001) and Noah Multi-parameterization (NoahMP; Niu et al. 2011). We
utilize NoahMP LSM because compared to Noah LSM it shows obvious improvements in
reproducing surface fluxes, skin temperature over dry periods, snow water equivalent, snow depth,
and runoff (Niu et al. 2011). Compared to LSM, one major advantage of WRF-Hydro system is
that, WRF-Hydro system can keep the infiltration capacity exceedance as ponded water within the
model domain. This ponded water is subsequently available for lateral redistribution, which
combine the ponded water with new precipitation for calculating the infiltration amount in the next
time step. WRF-Hydro has been tested in several different cases that focused on different
hydrometeorological forecasting and simulation problems (e.g., Gochis et al., 2015; Yucel et al.,
2015; Senatore et al., 2015; Arnault et al., 2016), and it shows reasonable accuracy in simulated
streamflow.
This study employs WRF-Hydro version 5 with a basic configuration. This configuration does not
use nudging technique as used in the National Water Model configuration and spatially distributed
soil-related parameters. The LSM is at a grid spacing of 3 km and the aggregation factor is 15, that



is, starting from a 3-km LSM resolution in the domain shown in Fig. 1, hydrological routing is
performed at a spatial resolution of 200 m. We use a time step of 10 seconds for the routing grid
to maintain model stability and prevent numerical dispersion of overland flood waves. The time
step also meets the Courant condition criteria for diffusive wave routing on a 200-m resolution
grid. The WRF-Hydro is configured to be offline or uncoupled mode ─ there is no online
interaction with WRF atmospheric model. Surface flow, saturated subsurface flow, gridded
channel routing, and a conceptual baseflow are active in this study. The gridded channel network
uses an explicit, one-dimensional, variable time-stepping diffusive wave. A direct output-equals-
input "pass-through" relationship is adopted here to estimate the baseflow. Although the baseflow
module is not physically explicit, it is very important because the water flow in the channel routing
are contributed by both overland flow and baseflow. If overland flow is active, it passes water
directly to the channel model. In this case the soil drainage is the only water resource flowing into
the baseflow buckets. If overland flow is deactivated but channel routing is still active, then WRF-
Hydro collects excess surface infiltration water from the land model, and passes this water into the
baseflow bucket. This bucket then contributes the water from both overland and soil drainage to
the channel flow. Therefore, the baseflow must be active if the overland flow is switched off. This
study does not consider lakes and reservoirs.
We use the geographic information system (GIS) tool that are developed by the WRF-Hydro team
to delineate the stream channel network, open water (i.e., lake, reservoir, and ocean) grid cells,
and groundwater/baseflow basins. Meteorological input for WRF-Hydro model system includes
hourly precipitation; near-surface air temperature, humidity, wind speed; incoming shortwave and
longwave radiation; and surface pressure. In this study, the hourly precipitation is from the
National Centers for Environmental Prediction (NCEP) Stage IV analysis at a spatial resolution of
4 km. The Stage IV data is based on combined radar and gauge data (Lin and Mitchell, 2005; Prat
and Nelson, 2015), and has been shown to be temporally well correlated with high-quality
measurements from individual gauges (see, e.g., Sapiano and Arkin, 2009; Prat and Nelson, 2015).
The other hourly meteorological input are from the second phase of the multi-institution North
American Land Data Assimilation System project, phase 2 (NLDAS-2) (Xia et al., 2012a,b), at a
spatial resolution of 12 km. NLDAS-2 is an offline data assimilation system featuring uncoupled
LSMs that are driven by observation-based atmospheric forcing.



**3 Calibration**
**3.1 Platforms**
We customized parallel PEST to work on two different workload managers and job schedulers:
SLURM at the National Energy Research Scientific Computing Center (NERSC), PBS at Argonne
National Laboratory Computing Resource Center, and Cobalt at Argonne Leadership Computing
Facility. The tests presented here are conducted on Edison of NERSC, which uses the SLURM
workload manager and job scheduler. Edison is a Cray XC30 with a peak performance of 2.57
petaflops per second, 133,824 compute cores, 357 terabytes of memory, and 7.56 petabytes of disk
storage. It has 5,586 nodes and 24 cores per node.
**3.2 PEST files**
Parallel PEST requires four types of input file:
1. Template files, which define the parameters to be calibrated. For example, we generated

13       CHANNEL.TPL, HYDRO.TPL, and GENPARAM.TPL based on the format of their

14       corresponding lookup tables, which are CHANNEL.TBL, HYDRO.TBL, and

15       GENPARAM.TBL, respectively. CHANNEL.TBL describes the features of a channel,

16       such as bottom width, channel side slope, and Manning's roughness coefficients.

17       HYDRO.TBL contains Manning's roughness coefficients for land-use types.

18       GENPARAM.TPL describes the parameters used in the Noah-MP LSM.

19    2. An instruction file, which defines the format of model-generated output files. For example,

20       WRF-Hydro can output time series of streamflow over the forecast points

21       (frxst_pts_out.txt) specified during model configuration. The instruction file follows the

22       format of frxst_pts_out.txt and specifies the line number of each calibrated forecast point

23       in frxst_pts_out.txt.

24    3. A control file, which supplies PEST with the size of the problem (e.g., how many

25       parameters to be calibrated; how many observational points); initial parameter values and

26       their lower and upper bounds; the increment of each parameter for forward-calculation; the

27       names of all template and instruction files; observational values, and weight for each

28       parameter to be calibrated. PEST requires all these three file types in both sequential and

29       parallel mode.



4. To run PEST in parallel mode, one also needs a management file to inform PEST where each slave's working folder is, as well as the names and paths of each model input file PEST must write (i.e., lookup tables that come from template files) and each model output file PEST must read (such as frsxt_pts_out.txt).

Parallel PEST uses a "master-slave" paradigm that starts model runs simultaneously in different folders (or by different "slaves"). The master of parallel PEST communicates with each of its slaves many times during the course of a calibration. When PEST needs to run a model in a particular folder, the master notifies the slave to start the model in that folder. Each slave starts the model execution accordingly, and informs the master that the model starts running. Once the simulation is completed in a particular folder, the slave signals the master, so the mater can read the particular output However, to the best of our knowledge, parallel PEST is not designed to run on HPCs directly. We developed scripts and an interface to enable parallel PEST to run on HPCs using SLURM, PBS, or Cobalt workload managers and job schedulers. This enables parallel PEST to run many slaves on the HPC; each slave runs a parallel code (such as WRF-Hydro) that uses more than one node, which could significantly increase the computational performance of model calibrations. Although this master-slave parallelism may not be as efficient as a fully MPI approach, it is sufficient for model calibration and requires the least effort for the current parallel PEST to run on HPCs.

## 3.3 Calibrated experiments

The primary objective of this study is to present the operational and the scale-up capability of the HPC-enabled parallel PEST for use with WRF-Hydro calibration over a relatively large domain. We focus less on extensively assessing the performance of the WRF-Hydro model. The calibration and validation is limited to only 7 days, considering it is long enough to achieve our objective and to understand WRF-Hydro's sensitivity to multiple parameters. The calibration compares WRF-Hydro modeled river discharge to U.S. Geological Survey (USGS) surface water observations. We originally choose 11 USGS sites across the study area. However, because of inaccuracies introduced when projecting geospatial data from one coordinate system to another by the ArcGIS tool, three of the observational sites were not properly assigned to the desired location on the channel network. This situation is common in hydrographic data processing and well known to



hydrologists (Sampson and Gochis, 2018). Among the remaining eight sites, four have
discontinuous or missing data over the calibration period. Therefore, we calibrated WRF-Hydro
using four USGS sites (referred to as Station 1, Station 2, Station 3, and Station 4 hereafter), as
shown in Fig. 1 with their site number. We then transfer the calibrated parameters to other sub-
basins in the study area to assess the transferability of the calibrated parameters. Although there
are many parameters, including spatially distributed parameters and constant parameters in the
lookup tables, that affect the model performance, we only calibrate the parameters in lookup tables
and do not consider the spatial variability of each parameter or their scaling factors. We
acknowledge that there are studies that calibrate a single scaling factor of overland roughness
coefficients (OVROUGHRTFAC) rather than the actual value of each land type in the lookup table
(e.g., Kerandi et al., 2018). Although this approach reduces the number of calibrated parameters,
it has less flexibility because changing one factor will change all the parameters that use the same
proportion.  In addition, a single scaling factor holds the same for the entire domain, which may
work well for a small domain, but could be problematic for a large domain. Thus, we suggest the
calibration of spatially distributed parameters requires more knowledge and understanding of the
study area and deserves future studies. In this study, we calibrate the roughness coefficients for
each land type rather than calibrating a single scaling factor.
For most calibration exercises we document here, the retention depth factor (RETDEPRTFAC) is
fixed at 0.001. This value is reasonable because the modelled discharge of our particular
configuration (Sect. 2.2) using default parameters is much lower than observed discharge.
Reducing this factor from 1 to 0.001 keeps less water in water ponds and more water on the surface
so it can contribute to river discharge. First, we calibrate 48 parameters based on a 3-day simulation
from April 9 to 12, 2013 (Table S1 in Supporting Information). We calibrate the Manning's
roughness coefficients for both channels and land-use types, the deep drainage (SLOPE),
infiltration-scaling parameter (REFKDT), and saturated soil lateral conductivity (REFDK). The
Manning's roughness coefficients control the hydrograph shape and the timing of the peaks; the
infiltration factor, saturated hydraulic conductivity, and deep drainage control the total water
volume. Second, based on the knowledge we learn from the 3-day calibration (see details in
Sect. 4.1), we redefine the number of parameters to calibrate and the range of many parameters
according to the literature (Soong et al., 2012) to maintain their physical meanings (Table 1). We





also extend our calibration period to 7 days to include a heavy precipitation period. Although a
period of 7 days is still very short compared to the traditional calibration period of at least 1 year,
we find it provides more appropriate parameter estimation—as well as better results of simulated
hydrograph shape and the total water volume—than does the 3-day calibration.
**3.4 Statistics**
This study employs three statistical criteria: Nash–Sutcliffe efficiency (NSE; Nash and Sutcliffe,
1970; Moriasi et al., 2007), root-mean-square error (RMSE), and Pearson correlation coefficient
(PCC). RMSE and PCC evaluate model performance in terms of bias and temporal variation. NSE
quantitatively describes the accuracy of modelled discharge compared to the mean of observed
data. Equation (1) calculates the NSE with defined variables:
$$NSE = 1 - \frac{\sum_{t=0}^{n}(Y_t^{obs} - Y_t^{sim})^2}{\sum_{t=0}^{n}(Y_t^{obs} - Y_{mean}^{sim})^2}, \quad (1)$$
where $Y_t^{obs}$ is the $t$th observed value from USGS sites for river discharge , $Y_t^{sim}$ is the $t$th
simulated value from the WRF-Hydro output, $Y_{mean}^{obs}$ is the temporal average of USGS observed
discharge, and $n$ is the total number of observation time points. An efficiency of 1 (NSE = 1)
corresponds to a perfect match between modeled discharge and observed data. An efficiency of 0
(NSE = 0) indicates that the model predictions are as accurate as the mean of the observed data.
An efficiency below zero (NSE < 0) occurs when the model is worse than the observed mean.
Essentially, the closer the NSE is to 1, the more accurate the model is.
**4 Results**
**4.1 Three-day calibration and validation**
Figure 2 shows the results of the 3-day modeled discharge (in cubic meters) using default and
calibrated parameters, as well as observed discharge from April 9 to 12. After the fifth iteration,
the difference in calibrated results between different iterations is relatively small, and PEST
performed 12 iterations before finding the optimum parameters. Here we only show results
generated by default parameters and by parameters calibrated from the 1st, 5th, and 12th iterations.
Over Stations 2, 3, and 4, which sit on rivers with relatively large water volumes, the discharge
modeled by the default parameters is much lower than discharge seen in observations. PEST





detects this underestimation. It immediately adjusts the parameters and increases the modeled
discharge during the first iteration. After adjusting the parameters for several iterations, the
modeled discharge gets much closer to the observations compared to the modeled results that used
the default parameters. Table 2 shows the statistics of model performance using default and
calibrated parameters for all four stations during the calibration and validation period. Compared
to the discharge that was modeled using the default WRF-Hydro parameters, overall, the calibrated
modeled discharge matches observations better, especially at the three stations with large volumes
of water. Note that this 3-day period only experienced a light rain over the study area. The
streamflow in the rivers is, therefore, mostly from groundwater and overland flow from upstream
or from previous precipitation events. The contribution of overland flow is small for this period
because the amount of precipitation was also small, so the main contributor to river discharge in
the real situation was from the groundwater. However, in this study WRF-Hydro uses a direct pass-
through groundwater model, which does not account for slow discharge and long-term storage of
the baseflow. Therefore, groundwater does not contribute much discharge to the channels. This
situation causes the model to greatly underestimate discharge, so the calibration adjusts critical
parameters aggressively to increase the streamflow to match the observations. When we apply
these calibrated parameters to the following days, the calibrated discharge is much higher than the
observed discharge during the heavy precipitation period, as shown by Fig. 3 and the RMSEs for
the validation period in Table 2. However, after the heavy precipitation event, the modeled
discharge decreases much faster than in the observed situation (Fig. 3). This again might be due to
the direct pass-through groundwater model we adopt in this study, which uses an output-equals-
input relationship between soil drainage and the discharge into river channels. This model does
not allow long-term storage of baseflow in each conceptual bucket, and thereby not be able to fully
represent the contribution of groundwater to streamflow.
From this 3-day calibration experiment, we learn that the WRF-Hydro output is not sensitive to
several parameters we calibrated in this particular study. For example, Manning's roughness
coefficients for several land types barely change during the calibration because these land types
(e.g., tundra, snow/ice) are not present in the study period and area. We also learn that even though
the calibrated WRF-Hydro parameters can generate discharge results that closely resemble
observations, the physical meaning of several parameters are not appropriate due to the wide range



of those parameters that we set in the PEST control file. For example, as shown in Table S1, the
Manning's roughness coefficient for stream order 1 (0.199) is calibrated smaller than that for
stream order 2 (0.218); the overland roughness coefficients for evergreen needleleaf forest (0.043)
and mixed forest (0.023) are calibrated smaller than cropland/woodland (0.046). Neither of these
is true in the real world.
**4.2 Seven-day calibration and validation**
Based on the knowledge we gained from the 3-day calibration, we adjust the range of critical
parameters in the PEST control file. For example, we set the Manning's roughness coefficient
larger for stream order 1 than for stream order 2. We also adjust the parameter range of the overland
roughness coefficient for multiple land covers, such as forests. With the adjusted range of
parameters, we perform 7-day calibration from 00:00 UTC on April 9, 2013, to 00:00 UTC on
April 16, 2013, when there is an increased streamflow that the simulation does not capture using
3-day calibrated parameters. The entire calibration takes 12 iterations. Figure 4 shows the results
of modeled discharge (in cubic meters) using default and calibrated parameters (from the first,
fifth, and 12th iterations), as well as observed discharge from April 9 to 16. Over Stations 2, 3, and
4, the modeled discharge using the default parameter underestimates the streamflow by more than
100%. PEST detects this underestimation and starts adjusting parameters to increase the discharge
to match the observations. Compared to the modeled discharge using default parameters during
the validation period, as shown in Table 3, the RMSE decreased from 624.9 (Station 1), 5162.9
(Station 2), 4990.0 (Station 3), and 5098.3 m$^3$/sec (Station 4) to 283.1, 637.9, 666.8, and 1202.8
m$^3$/sec, respectively. The correlation coefficient between observed and modeled discharge
increased from 0.71, 0.90, 0.87, and 0.82 to 0.97, 0.99, 0.96, and 0.86. Note that, although the
calibration helps three stations (Station 2, 3, and 4) with large water volumes to generate more
reasonable results than the default parameters, the results for Station 1, which has a relatively small
volume of water, is not always better than the discharge that is modeled using default parameters
(Tables 2 and 3). This might be because we use the same absolute weight for all the stations when
we perform the calibration. Using a higher weight for Station 1 may help improve this situation
and generate better results for this station.



Although a period of 7 days is still very short for calibration compared to traditional calibration
period of at least 1 year, we find that the 7-day period provides better and more appropriate
parameter estimation than does the 3-day calibration, and it does a better job of capturing the
hydrograph shape and the total water volume. Comparing the validation statistics between Tables
2 and 3 as well as Figs. 3 and 5, we find the 7-day calibrated parameters generate better results
than do the 3-day calibrated parameters for both water volume and the hydrograph shape over the
validation period. Compared to the discharge modeled using 3-day calibrated optimum parameters,
there is a 17–33% increase in the simulated streamflow (1,400–2,800 $m^3$/sec) calculated over
Station 2, 3, and 4 using the 7-day calibrated parameters from April 12 to 16. The RMSE is 3,400–
3,600 $m^3$/sec when calculated using the 3-day calibration, but this decreases to 600–1,200 $m^3$/sec
when calculated using the 7-day calibration. The correlation coefficient between observed and
modeled discharge using the 7-day calibration is 0.8–0.99, but that using the 3-day calibration is
only 0.7–0.8. However, there is still a problem with the temporal variability of the modeled
discharge, especially over the rivers with large discharge. When there is precipitation, the
discharge immediately increases and is higher than the observed discharge. After the precipitation
period, when the observed discharge still stays high, the modeled discharge decreases sooner and
thus is smaller than the observed discharge. This might be the direct pass-through approach
simplified groundwater flow, and does not represent the interaction between stream flow and
groundwater properly in this case study.
**4.3 Evaluation of spatial transferability of the modeling system**
In this section, we apply the calibrated parameters for the four stations (black circles) in Fig. 1 to
other 13 stations in the study area. As mentioned before, because of inaccuracies in the spatial
location of station data and digital elevation models, and because of small errors introduced when
projecting geospatial data from one coordinate system to another using the ArcGIS tool, only four
stations are mapped on the river systems (crosses in Fig. 1); others are slightly shifted out of their
closest grid cell. One of these four sites (Station 5) is located on a relatively small river, and others
are located on larger rivers. The following analyses assess the transferability of the calibrated
parameters from particular sites to other sites that are in the study area but not calibrated. The
assessment compares the observed discharge with the closest grid cells from the discharge output
of WRF-Hydro. Figure 6 shows the observed and modeled discharge using default and calibrated





parameters. Overall, WRF-Hydro's default parameters underestimate the discharge. WRF-Hydro also generates an earlier discharge peak compared to observations over the four stations (Stations 5, 6, 7, and 8) in this particular study. The calibrated model results increase the discharge and generate a hydrograph shape that is closer to the observations than the default model results do. The absolute error of simulated discharge decreases by 12.8%, 22.8%, 46.8%, and 49.9%, respectively, over Stations 5 through 8, compared to the default simulated discharge. However, because we did not specifically calibrate these stations based on observations, there are still differences between the calibrated results and observations.

# 5 Discussion and summary

## 5.1 Scale-up capabilities

The ability to scale up calibration of WRF-Hydro using parallel PEST on HPCs is determined by two factors: the scale-up capability of WRF-Hydro, and the scale-up capability of PEST. In the course of calibrating WRF-Hydro, PEST must run the WRF-Hydro model many times. PEST makes some model runs to calculate Jacobian matrix (Doherty, 2016). These model runs are independent between slaves. Each slave run the model using temporarily incremented parameters that are defined in the template and control files. These model runs can be easily parallelized. However, PEST also need to make some other model runs to test parameter upgrades. These runs are calculated based on different Marquardt lambdas. The search for a Marquardt lambda that achieves the best set of parameters is a serial procedure — what lambda to use next depends on the outcome of the model run conducted using the previously chosen lambda. This in fact is the major bottleneck of parallelization of the PEST code. Although serial testing of Marquardt lambdas may quickly find the optimal Marquardt lambda in the first or second series of model runs, it is an inefficient use of computing resources because other processors are idle while only one process is searching the lambdas. This is especially true when the model domain is large and requires extensive computing resources.

This study employs "partial parallelization" for the lambda-testing procedure (Doherty, 2016), so all the processors can be used to calculate parameter upgrades based on a series of lambda values that are related to each other by a factor of RLAMFAC set in the PEST control file. This partial





parallelization makes the scale up challenging when more processors are in use, because
generating many Marquardt lambdas does not always guarantee that the best Marquardt lambdas
were the ones generated. As a result, the calibration process may converge more slowly when
using more slaves than it does when using less slaves. We tested different numbers of slaves (35,
50, 70, and 105) for the 32-parameter calibration experiment. In total, each of these tests uses 71,
101, 141, and 211 nodes; two nodes for each slave run WRF-Hydro, and one node runs PEST
master to coordinate jobs and communicate with the slaves. The results shown in Figs. 2–6 and
Tables 2–3 are from a calibration using 35 slaves; PEST conducted 12 iterations before finding
the optimum parameters. We find that using different numbers of slaves generates slightly different
parameter values and involves different numbers of iterations. For example, using 70 slaves only
takes eight iterations and 41% of the wall-clock time 35 slaves used to find the optimum
parameters. The calibrated parameters are slightly different from those generated by 35 slaves
(Table 1, last column), and they generate slightly better results than does the 35-slave test
compared to observed discharge. We finish the 12 iterations using 35 slaves (71 nodes) within 73
hours, and the eight iterations using 70 slaves (141 nodes) within 30 hours. More than 800 model
runs were conducted for entire calibration process including calculating the Jacobian Matrix as
well as testing the parameter upgrades. In fact if more nodes were used by each slave for the
calculation, the wall-clock time can be further reduced. If these calibration were conducted
sequentially on personal computers, the same calibration process would have taken 60–80 days for
a 7-day calibration.
Our study finds that, depending on the number of parameters being calibrated (e.g., 32 parameters
in this particular study), using 32 to 64 slaves shows fairly good scale-up capability; most of the
time consumed by PEST is for running WRF-Hydro, and the number of slaves can be used to carry
out model runs to generate the Jacobian matrix. However, using 105 slaves (211 nodes) does not
result in fewer iterations or a shorter wall-clock time than using 70 slaves. In fact, using more than
64 slaves may not be necessary because generating many Marquardt lambdas does not always
guarantee generating the best Marquardt lambdas. In addition, at least for the calibrations
conducted in this study, in each iteration, PEST runs the model either 32 or 64 times to calculate
the Jacobin matrix. Therefore, having more than 64 slaves would most likely render some slaves
idle and is not an efficient use of computing resources.



## 5.2 Summary

WRF-Hydro is a new, and perhaps the first practical, computer code that can run on HPCs and can model the entire hydrological cycle using physics-based sub-models and very high-resolution input datasets (e.g., radar). The hydrological community has desired this capability for decades, although it requires intensive computing resources. Thus, the calibration of this model would ideally be conducted on HPCs in parallel as well, especially when the model covers a large domain rather than the basin scale. This study ports an independent model calibration tool, parallel PEST, to HPC clusters and links it to WRF-Hydro to help WRF-Hydro users calibrate the model within a much shorter wall-clock period. This tool's uniqueness lies in its flexibility and robustness to calibration any parameters in WRF-Hydro. It is also unique in its use of two levels of parallelization across many slaves running PEST, with each slave running a simulation of WRF-Hydro. The calibration tool presented in this study also applies to any other hydrological models and similar earth system models that use parameterization to model physics. We present the optimum parameters identified from the calibration of this particular study case and area, but the calibrated parameters can be significantly different if one uses different physics, such as exponential storage-discharge function for a groundwater model, or reach-based channel routing.

We apply the HPC-enabled parallel PEST to WRF-Hydro to investigated a major flood event that occurred over the Midwestern United States in April 2013. Our precipitation inputs were derived from a radar, gauge, and satellite rainfall product named Stage IV. The calibrated parameters include Manning's roughness coefficients for both channels and land-use types, deep drainage, the infiltration scaling parameter, and saturated soil lateral conductivity. We evaluated the performance of WRF-Hydro in hydrograph features such as volume and timing of flood events. We also assessed the spatial transferability of the calibrated parameters in the study area.

The following are the primary findings of this study:

1. For this particular study, the HPC-enabled PEST calibration tool can speed up WRF-Hydro calibration by a factor of 30, compared to a serial calibration procedure.



2. Calibrated WRF-Hydro improves the modeled hydrographs compared to the default model results. The RMSE of discharge over the three large stations are reduced by 76–86% with calibration for the validation period.

3. Although the calibration period in this study is relatively short, we found that the longer the calibration period is, the better the model results are when compared to observations. It is difficult to precisely define what length of data is sufficient to identify model parameters so that they can also be used for other periods, because different models have different levels of complexity and different catchments have different information content in each year of hydrological record. Because of the heavy computation load for the calibration of WRF-Hydro, it would be challenging to calibrate yearlong time series.

4. Although there are inaccuracies when mapping the USGS stations onto the river systems, we found the WRF-Hydro calibrated parameters are helpful for nearby locations. The absolute bias over the four assessed stations decreased by 12–50% as a result of using the calibrated parameters, compared to using the default parameters for their simulations. The mapping issue are often random, or non-systematic; there is no generalizable way to automate the correction procedure with a high degree of fidelity. Manual manipulation or specification of the data is often required (Sampson and Gochis 2018).

5. Using different groundwater models can generate very different results and will require a completely different set of parameters for WRF-Hydro to model the observed discharge. Our preliminary testing shows that, using exponential storage-discharge function with the default parameters provided by WRF-Hydro, the modeled discharge was larger than observations. Thus, the calibration will need to adjust the parameters to reduce the discharge.

*Data and Code availability.* The observed river discharge is downloaded from the USGS Surface-Water Data website, available at https://waterdata.usgs.gov/nwis/sw. The Stage IV precipitation data were downloaded from https://data.eol.ucar.edu/dataset/21.093. PEST was downloaded from http://www.pesthomepage.org/Downloads.php. We use the Unix PEST version 13.6. The scripts and files that are developed in this study and required by PEST for calibrating WRF-Hydro are available at https://www.zenodo.org/record/1490230#.W_XI6TFRdhE. DOI: 10.5281/zenodo.1490230.





*Author contributions.* JW proposed the project and developed the study case in WRF and WRF-
Hydro. CW developed the scripts/code to port the parallel PEST to DOE supercomputers and adapt
it to work with WRF-Hydro. AO operated the ArcGIS tool to delineate the high resolution grid
cells to include stream channel network, open water and groundwater/baseflow basins. RK provide
high-level guidance and insight for the entire project. All authors commented on this manuscript.
*Competing interests.* The authors declare that they have no conflict of interest
*Acknowledgements.* This work is supported under a Laboratory Directed Research and
Development (LDRD) Program at Argonne National Laboratory, through U.S. Department of
Energy (DOE) contract DE-AC02-06CH11357. Computational resources are provided by the
DOE-supported National Energy Research Scientific Computing Center, Argonne National
Laboratory Computing Resource Center, and Argonne Leadership Computing Facility. Our special
thanks to PEST developer John Doherty and the entire WRF-Hydro team, especially Kevin
Sampson, for his guidance on the ArcGIS tool.

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

**Figure captions:**





**Figure 1:  Eight USGS sites over the study area. The four circles are sites that are used for**
**calibrations; the four crosses are sites that are used for transferability assessment. USGS site**
**numbers corresponding to the site index used in this study are listed on the top left corner of**
**the map.**
**Figure 2: Observed and modeled discharge (m³/sec) using default and calibrated parameters**
**during a 3-day calibration period (April 9–12, 2013) over the four stations indicated by the**
**black circles in Fig. 1.**
**Figure 3: Observed and modeled discharge (m³/sec) using optimum parameters identified**
**from a 3-day calibration during a validation period (April 13–24, 2013) over the four stations**
**indicated by black circles in Fig. 1.**
**Figure 4: Same as Fig. 2, but during a 7-day calibration period (April 9–15, 2013).**
**Figure 5: Same as Fig. 3, but the modeled discharge uses 7-day calibrated parameters over**
**a validation period from April 17 to 24, 2013.**
**Figure 6: Observed and modeled discharge (m³/sec) using default and the optimum**
**parameters identified by a 7-day calibration over four stations that are in the study area**
**(indicated by crosses in Fig. 1).**



1 **Table 1: 7-day calibrated parameters and the optimum parameter found by the calibration.**

| Calibrated parameters | Default | Lower bound | Upper bound | Optimum parameters 35 slaves (12[a]) | Optimum parameters 70 slaves (8[a]) |
|---|---|---|---|---|---|
| mannn1 | 0.55 | 0.35 | 0.6 | 0.440658 | 0.469277 |
| mannn2 | 0.35 | 0.15 | 0.35 | 0.35 | 0.209592 |
| mannn3 | 0.15 | 0.08 | 0.15 | 8.00E-02 | 8.00E-02 |
| mannn4 | 0.1 | 0.05 | 0.15 | 0.15 | 0.15 |
| mannn5 | 7.00E-02 | 0.02 | 0.1 | 6.13E-02 | 8.59E-02 |
| mannn6 | 5.00E-02 | 0.015 | 0.1 | 1.50E-02 | 1.50E-02 |
| mannn7 | 4.00E-02 | 0.01 | 0.08 | 1.41E-02 | 1.36E-02 |
| mannn8 | 3.00E-02 | 0.005 | 0.06 | 5.00E-03 | 1.12E-02 |
| mannn9 | 2.00E-02 | 0.003 | 0.05 | 3.00E-03 | 5.00E-03 |
| mannn10 | 1.00E-02 | 0.0001 | 0.03 | 4.09E-04 | 1.00E-04 |
| xslope1 | 0.1 | 1.00E-04 | 1 | 0.64603 | 0.663051 |
| refdk | 2.00E-06 | 1.00E-08 | 1.00E-05 | 9.85E-08 | 1.83E-07 |
| refkdt | 1 | 0.01 | 5 | 4.98912 | 1.49769 |
| ovn1 | 2.50E-02 | 0.005 | 0.06 | 2.12E-02 | 5.00E-03 |
| ovn2 | 3.50E-02 | 0.015 | 0.06 | 6.00E-02 | 6.00E-02 |
| ovn3 | 3.50E-02 | 0.015 | 0.06 | 5.45E-02 | 6.00E-02 |
| ovn4 | 5.50E-02 | 0.015 | 0.1 | 5.50E-02 | 5.50E-02 |
| ovn5 | 3.50E-02 | 0.015 | 0.06 | 6.00E-02 | 6.00E-02 |
| ovn6 | 6.80E-02 | 0.035 | 0.25 | 3.50E-02 | 0.144637 |
| ovn7 | 5.50E-02 | 0.015 | 0.25 | 1.50E-02 | 1.75E-02 |
| ovn8 | 5.50E-02 | 0.015 | 0.25 | 5.50E-02 | 5.50E-02 |
| ovn9 | 5.50E-02 | 0.015 | 0.25 | 5.50E-02 | 5.50E-02 |
| ovn10 | 5.50E-02 | 0.015 | 0.3 | 1.50E-02 | 1.50E-02 |
| ovn11 | 0.2 | 0.1 | 0.3 | 0.27072 | 0.3 |
| ovn12 | 0.2 | 0.1 | 0.3 | 0.1 | 0.20025 |
| ovn13 | 0.2 | 0.1 | 0.3 | 0.1 | 0.20025 |
| ovn14 | 0.2 | 0.1 | 0.3 | 0.291836 | 0.1 |
| ovn15 | 0.2 | 0.1 | 0.3 | 0.1 | 0.1 |
| ovn16 | 5.00E-03 | 0.001 | 0.01 | 1.00E-03 | 1.00E-03 |
| ovn17 | 7.00E-02 | 0.005 | 0.1 | 7.27E-02 | 7.01E-02 |
| ovn18 | 7.00E-02 | 0.005 | 0.1 | 7.27E-02 | 7.01E-02 |
| ovn19 | 3.50E-02 | 0.015 | 0.06 | 4.04E-02 | 3.52E-02 |

2   [a]  Number of iterations during the calibration before finding the optimum parameter.





**Table 2: Statistics of model performance using optimum and default (in the parenthesis)**
**parameters for Stations 1-4 during the calibration and validation period.[a]**

| Statistics | Station 1 | Station 2 | Station 3 | Station 4 |
|---|---|---|---|---|
| Calibration | | | | |
| NSE | 0.14 (0.73) | **0.70 (-54.4)** | **0.49 (-157.3)** | **-3.7 (-1316.9)** |
| RMSE | 123.6 (69.7) | **292.7 (3944.7)** | **227.2 (4001.1)** | **263.7 (4413.2)** |
| PCC | **0.97 (0.91)** | 0.90 (0.92) | **0.94 (0.87)** | **0.83 (0.66)** |
| Validation | | | | |
| NSE | **0.83 (0.39)** | **-0.51 (-4.2)** | **-0.81 (-46.5)** | **-1.02 (-5.54)** |
| RMSE | **276.9 (528.3)** | **3452.1 (6410.9)** | **3416.8 (17503.4)** | **3694.5 (6655.6)** |
| PCC | **0.96 (0.8)** | **0.80 (0.76)** | **0.76 (0.21)** | 0.72 (0.72) |

[a] The calibration period is 3 days (April 9–12) and includes 48 parameters. The validation period
is April 13–23. Bold typeface indicates the calibrated model results are closer to observations
compared to the default model results. NSE and PCC are unitless; RMSE is in $m^3$/sec.





**Table 3: Same as Table 2, but for 7-day calibration.[a]**

| Statistics | Station 1 | Station 2 | Station 3 | Station 4 |
|---|---|---|---|---|
| | | Calibration | | |
| NSE | -0.53 (-0.19) | **0.99 (-4.6)** | **0.98 (0.12)** | **0.92 (0.1)** |
| RMSE | 217.3 (191.2) | **241 (5162.9)** | **273.6 (4990.9)** | **607.8 (5098.3)** |
| COR | **0.83 (0.64)** | **0.99 (0.90)** | **0.96 (0.87)** | **0.86 (0.82)** |
| | | Validation | | |
| NSE | **0.86 (0.30)** | **0.91 (-4.6)** | **0.9 (-4.1)** | **-0.69 (-4.6)** |
| RMSE | **283.1 (624.9)** | **637.9 (5162.9)** | **666.8 (4990.9)** | **1202.8 (5098.3)** |
| COR | **0.97 (0.71)** | **0.99 (0.90)** | **0.96 (0.87)** | **0.86 (0.82)** |

[a] The calibration period is 7 days (April 9–12) and includes 48 parameters. The validation period
is April 13–23. Bold typeface indicates the calibrated model results are closer to observations
compared to the default model results. NSE and COR are unitless; RMSE is in $m^3$/sec.





**Figure 1: Eight USGS sites over the study area. The four circles are sites that are used for calibrations; the four crosses are sites that are used for transferability assessment. USGS site numbers corresponding to the site index used in this study are listed on the top left corner of the map.**





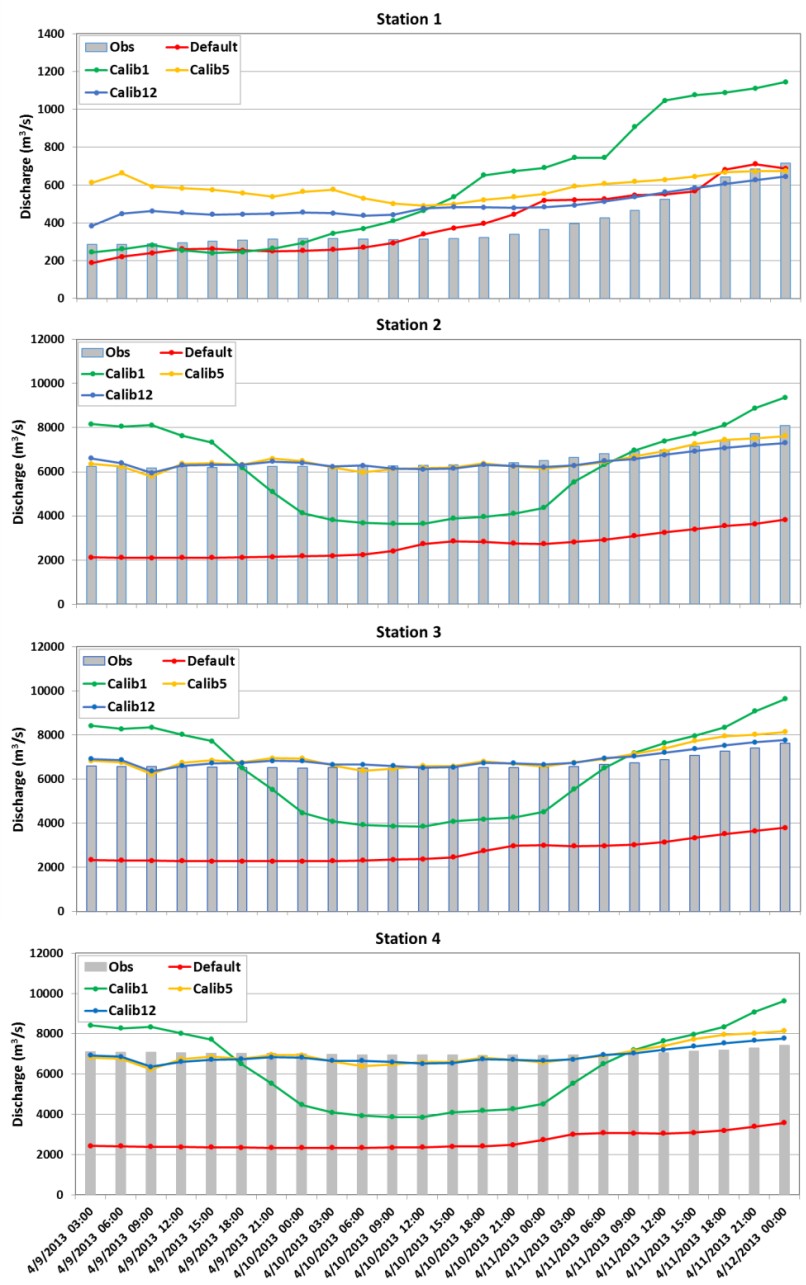

**Figure 2: Observed and modeled discharge (m³/sec) using default and calibrated parameters**

**during a 3-day calibration period (April 9–12, 2013) over the four stations indicated by the**

**black circles in Fig. 1.**





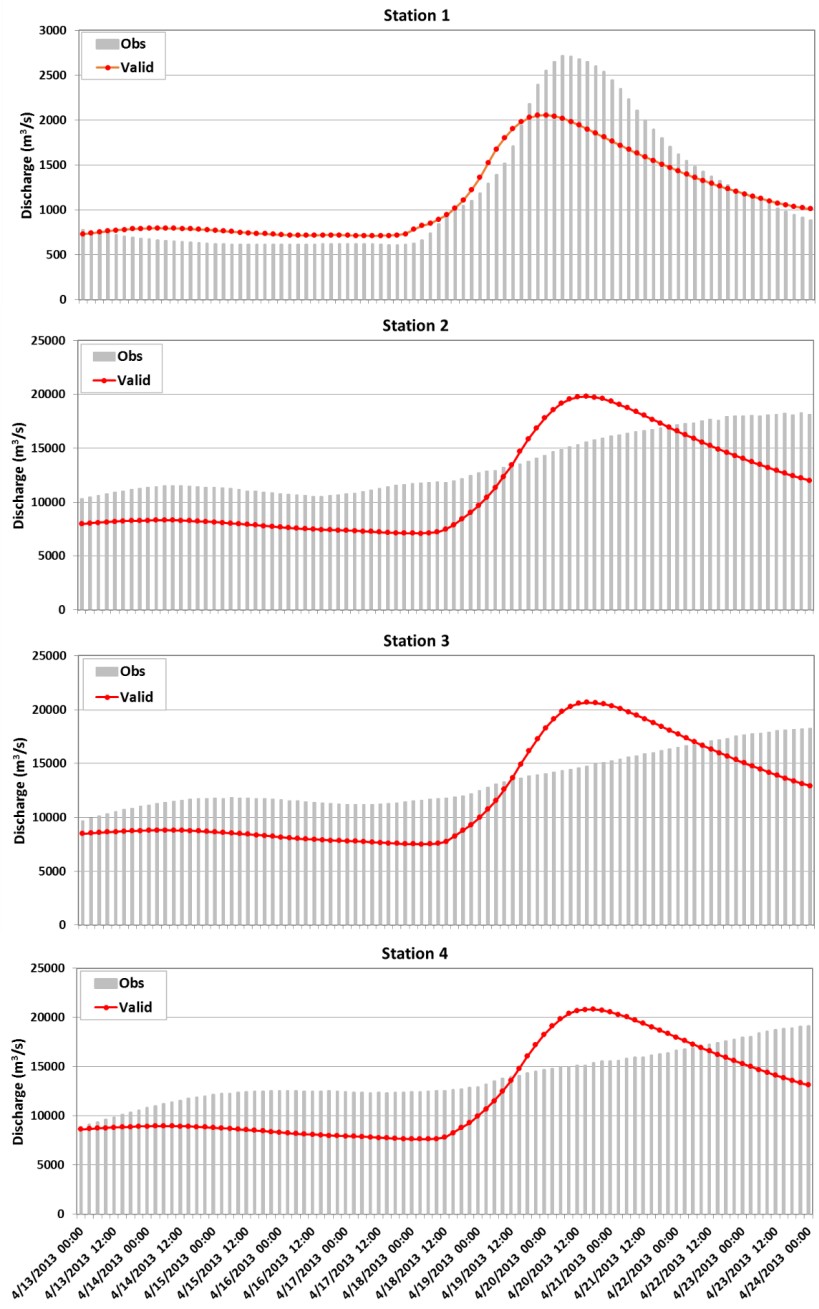

**Figure 3: Observed and modeled discharge ($m^3$/sec) using optimum parameters identified**

**from a 3-day calibration during a validation period (April 13–24, 2013) over the four stations**

**indicated by black circles in Fig. 1.**





2    **Figure 4: Same as Fig. 2, but during a 7-day calibration period (April 9–15, 2013).**





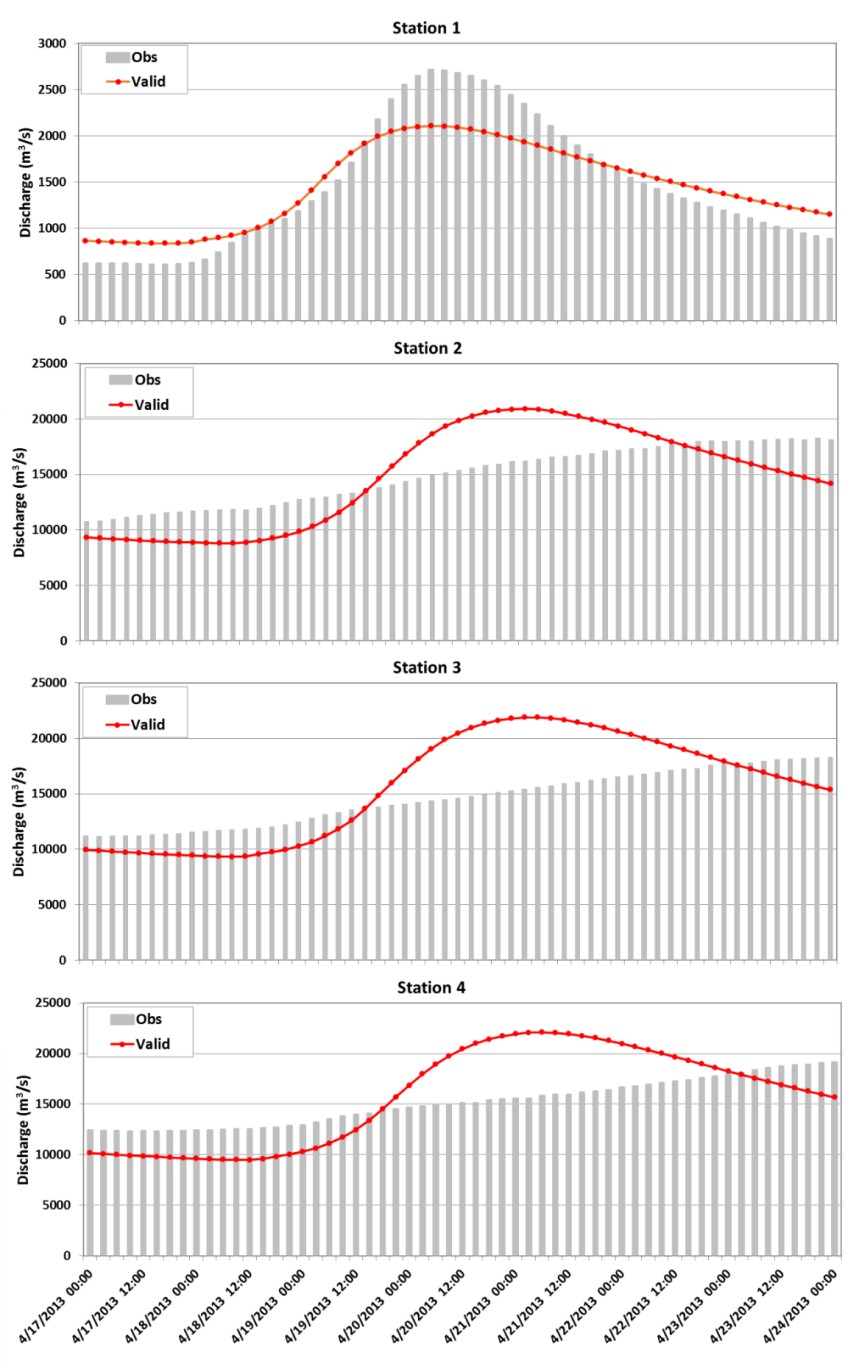

**Figure 5: Same as Fig. 3, but the modeled discharge uses 7-day calibrated parameters over**

**a validation period from April 17 to 24, 2013.**





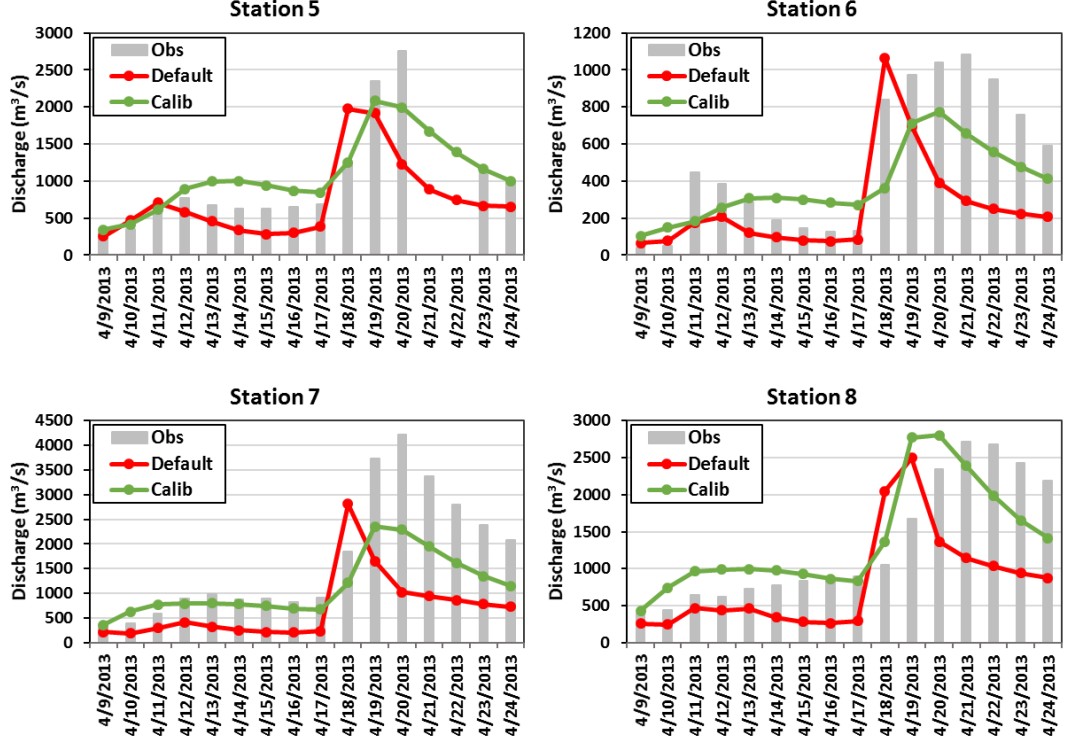

**Figure 6: Observed and modeled discharge (m³/sec) using default and the optimum**

**parameters identified by a 7-day calibration over four stations that are in the study area**

**(indicated by crosses in Fig. 1).**