# Peer review of "A parallel workflow implementation for PEST version 13.6 in"

_Geoscientific Model Development, 2018_

## Referee Comment (RC1) · Doherty (Referee) · 6 Dec 2018

The authors describe modifications that they made to PEST to enhance its use on a HPC. They then describe use of their modified PEST in calibration of a complex surface water model.

While I found the paper interesting, I found that it was lacking in information in some respects. For example nothing is said about the interface that they constructed between parallel PEST and the run management software that they employed. Nor was

any reference made to PEST settings. While I agree with the authors that use of inversion methods that can parallelize model runs and handle the estimation of many parameters employed by a complex model is a much-needed addition to the arsenal of surface water modelling, I think that many more advances could be made than the authors have made. In particular, there was no mention of the use of Tikhonov regularisation to accommodate parameter nonuniqueness at the same time as it promulgates uniqueness through obtaining a set of parameters that "make sense" from an expert knowledge point of view. This, I think, is one of the strongest arguments for use of gradient-based, highly parameterized methods in regional surface or land use model calibration, that is the ability to not just accommodate nonuniqueness, but to turn the "wiggle room" engendered by nonuniqueness into formulation of an inverse problem that can actually make regionalization and transportability of parameters a reality.

The authors use a simple objective function. This may be ok for some inverse problems. However as they point out, some of the smaller flows (in terms of location in space and location in a single flow time series) are not as well fitted as they could be. Perhaps weights should be a function of flow – and of location. Perhaps other important aspects of the flow time series should be made more visible to PEST through formulation of separate, targetted objective function components to ensure that these aspects of the time series are also well fit.

The authors make a big deal out of their modifications to parallel PEST so that it is HPC-friendly. Actually, I think that the BEOPEST version of PEST has similar capabilities. The original version of BEOPEST used both MPI and TCP/IP for communication between master and slaves (now called manager and workers). Now only TCP/IP is used. One of the reasons that BEOPEST's capabilities exceed those of parallel PEST in the HPC environment (actually on any network) is that the manager does not need to write model input files and read model output files across the network. This makes run management must faster, more secure, and able to take place in a greater variety of network environments.

In summary, I think that what the authors have done is good. However I also think that the potential for regional surface water model calibration and uncertainty analysis in a HPC environment still remains largely untapped. Some of this potential will be realised with use of singular value decomposition to ensure numerical stability when inverse problems are ill-posed, use of Tikhonov regularisation to ensure parameter sensibility and transportability under the same conditions, and more creative formulation of the objective function than the authors have done.

---

## Referee Comment (RC2) · Anonymous Referee #2 · 16 Jan 2019

The paper of Wang et al. deals with a potentially interesting implementation of the parallel version of the PEST software. PEST is a powerful and very useful tool for hydrologists, helping them during long and "exhausting" calibration sessions. Therefore, introducing the portability of parallel PEST to HPCs is good news and, specifically for the present paper, the main theme to highlight. Nevertheless, in my opinion the way the paper is structured mainly highlights, instead of the advantages of the novelty, the performances of the PEST calibration, which is something widely and well assessed by the hydrology research community. Almost all figures and tables deal with PEST

results. Furthermore, the calibration procedure presented is questionable from different points of view (some of which are exposed later). The most interesting/innovative Section of the paper is Section 5.1, but the analysis of scale-up capabilities should be described with much more detail. Concerning the main outcomes of the paper highlighted in the summary, points from 2 to 5 are quite obvious (they deal with the recognized skills of the PEST software), while point 1 should be expanded: what does a factor of 30 "with respect to a serial calibration" exactly mean? In my opinion it's not a rigorous statement. What do the authors exactly mean with "serial"? Even though PEST calibration is serial, WRF-Hydro can run in a parallel fashion, and the speed of the calibration process would depend on the number of nodes used for the hydrological simulation. A possible idea is to provide hints about the trade-off between the number of nodes/CPUs used for running the parallel model (i.e., WRF-Hydro in this case) and the number of nodes/CPUs used for running PEST in a parallel fashion. I guess it depends somehow also on the dimensions of the domain (and no information is given here about the number of cells in which the basin is discretized, so the reader has no idea about the actual computational burden).

Another important point, that should be better discussed, is the missed capability of the implemented version of PEST to deal with the calibration of spatially distributed parameters. This is important because it's reasonable to expect parallel PEST executions with WRF-Hydro for wide domains, and wide domains often need spatial differentiation of spatially distributed parameters, like, e.g., OVROUGHRTFAC, RETDEPRTFAC or other spatially distributed parameters available with WRF-Hydro v5.0. By the way, another limitation is that, at least as I understand, the calibration is available only against observed streamflow. Of course, this is the first option but not the unique one (one can decide to calibrate also against, e.g., soil moisture or latent heat flux data).

Finally, another important point is to (at least) discuss the problem of equifinality, which is incidentally (but not explicitly) dealt with in P11 L29 – P12 L5.

Summarizing, though I acknowledge that the research presented is potentially interesting and innovative, I suggest to re-think the paper highlighting much more the computational benefits provided and reviewing the calibration performed in the case study. Following, a (not comprehensive) list of doubts regarding the calibration procedure and other minor comments and typos. I hope my comments can help improving the research.

Doubts about the calibration procedure:

Even though I acknowledge that authors decided to "focus less on extensively assessing the performance of the WRF-Hydro model", several aspects of the calibration procedure are very questionable.

1. no information about spin-up. This is extremely important, especially for such a short range calibration (only few days). The model should be run in advance (at least one month, I would say) in order to let several variables (e.g., moisture fields) have a realistic spatial distribution.

2. the authors state that: April 8-11 moderate rain, April 12-14 no rain, April 15-18 rain, peak flow April 19. 3-day calibration is: April 9-11 (to be precise, April 12 at midnight), then validation is April 13-23 (April 12 is missed). 7-days calibration is April 9-15, validation is April 17-23. To me, it does not make too much sense that 4 more days are added when only the last one is rainy. It would be much better to calibrate the model with respect to a previous flood event, as it is usual. After all, observing graphs in figures 3 and 5 one after another just shows that increasing the number of days used for calibration improves the performances (but this is rather obvious), even though not yet enough.

3. In order to deal with the observed streamflow in Section 1, it is fundamental to work with weights.

Minor comments, grammar and typos

P6 LL6-17: not clear if in this case overland flow is switched on. It should.

P6 L19: probably "tools"

P7 L18: GENPARM.TBL

P8 LL11-12: master, not mater. The full stop is missing.

P8 L30: As it is a common problem, it is usually solved 'simply' reallocating manually the stations. It's a pity to miss streamflow data for this reason

P9 L24 and following: I suggest to explicitly declare also the meaning of the ovn parameter

P12 L17: 50%, maybe

Figs.2 and 3: April 12 is missing. It should be the first validation day, I guess.

Figs. 4 and 5: the same for April 16

Table 3: the note is incorrect, it refers to information about the 3-day calibration

Section 4.3: this is a purely "hydrological" analysis that could be skipped, given the numerous limitations of the calibration procedure and the focus on the implementation of the PEST software

P16 LL9-10: please check the sentence

P16 L18: to investigate

―――――――――――――――――

---

## Author Comment (AC1) · 9 Mar 2019

We would like to express our deep appreciation to the two reviewers (Dr. Doherty and the anonymous reviewer) for the thoughtful comments and insightful suggestions. Working to resolve these comments helps us add lots of interesting science and add value to the manuscript.

The primary objective of this study, as pointed out by Reviewer #2, is to build a bridge for linking the parallel PEST and WRF-hydro on the basis of HPC clusters and explore

the computational benefits of this bridge. We do not attempt to extensively assess each individual tool or address questions in each individual domain, such as optimizing the objective functions in PEST or calibrating WRF-Hydro to achieve the best set of model parameters. However, we appreciate the opportunity every much during the revision of this manuscript by learning more about PEST especially the method of regularization for calibrating environmental models. We are also very glad that the reviewers found the bridge we built useful for helping WRF-Hydro users with the long and tedious model calibration.

In the revised version of this manuscript, several major changes are made based on both reviewers' comments/suggestions. They are listed below:

1. We re-do the WRF-Hydro calibration using SVD-based regularization method in PEST.

2. We consider prior information for the calibrated parameters.

3. We also consider different weight for the stations that are calibrated, based on their inversed mean of discharges.

4. To test the computational benefits of the bridge, we design five experiments by assigning different amount of computing resource for parallel PEST and for parallel WRF-hydro.

5. To constrain the problem size due to the limits of computing resource, we reduce the number of calibrated parameters to 22 according to the model sensitiveness of this particular study.

Please find our one-on-one response in Supplement to each reviewer's comment. A complete list of the changes made for the revised manuscript can be found in the "track changes" version of the manuscript. A clean version of the revised manuscript is also attached at the end.

Sincerely,

[Figure]

Jiali Wang jialiwang@anl.gov

Please also note the supplement to this comment:
https://www.geosci-model-dev-discuss.net/gmd-2018-253/gmd-2018-253-AC1-
supplement.pdf

**Supplement:**

We would like to express our deep appreciation to the two reviewers (Dr. Doherty and the anonymous reviewer) for the thoughtful comments and insightful suggestions. Working to resolve these comments helps us add lots of interesting science and add value to the manuscript.

The primary objective of this study, as pointed out by Reviewer #2, is to build a bridge for linking the parallel PEST and WRF-hydro on the basis of HPC clusters and explore the computational benefits of this bridge. We do not attempt to extensively assess each individual tool or address questions in each individual domain, such as optimizing the objective functions in PEST or calibrating WRF-Hydro to achieve the best set of model parameters. However, we appreciate the opportunity every much during the revision of this manuscript by learning more about PEST especially the method of regularization for calibrating environmental models. We are also very glad that the reviewers found the bridge we built useful for helping WRF-Hydro users with the long and tedious model calibration.

In the revised version of this manuscript, several major changes are made based on both reviewers' comments/suggestions. They are listed below:
1. We re-do the WRF-Hydro calibration using SVD-based regularization method in PEST.
2. We consider prior information for the calibrated parameters.
3. We also consider different weight for the stations that are calibrated, based on their inversed mean of discharges.
4. To test the computational benefits of the bridge, we design five experiments by assigning different amount of computing resource for parallel PEST and for parallel WRF-hydro.
5. To constrain the problem size due to the limits of computing resource, we reduce the number of calibrated parameters to 22 according to the model sensitiveness of this particular study.

Please find our one-on-one response below to each reviewer's comment. A complete list of the changes made for the revised manuscript can be found in the "track changes" version of the manuscript. A clean version of the revised manuscript is also attached at the end.

Sincerely,
Jiali Wang
jialiwang@anl.gov

**Reviewer #1**
The authors describe modifications that they made to PEST to enhance its use on a HPC. They then describe use of their modified PEST in calibration of a complex surface water model. While I found the paper interesting, I found that it was lacking in information in some respects. For example nothing is said about the interface that they constructed between parallel PEST and the run management software that they employed.

*Response:*

Thank you for your comment. The interface is the most important thing we built in this study, and testing the operational feasibility and computational benefit of this interface are the main objectives of this manuscript. Hence it definitely should be described as you suggested. We add this paragraph in **Section 3.2 PEST files and settings**:

"The interface we have built between parallel PEST and the management software (SLURM here) is, in general, used for (1) setting the number of workers and the nodes for each worker to conduct a model run (WRF-Hydro here); (2) finding the nodes that are available; (3) setting up the working directory for the workers; (4) identifying the nodes that work for each worker; (5) passing the global files (same for all the working directory) to all the workers (these files include the lookup table files that are not to be calibrated, the namelist files for both LSM and hydrological sector, and restart files that generated by the previous simulations, or spin-up period); and (6) submitting the job for the entire calibration process, including parallel PEST and parallel WRF-hydro. This job can be submitted as a cold-start run or as a restart. The main difference for this interface on different management software is that different management software has its own way to submit jobs and identify available nodes. This difference requires some changes in the script we developed."

Nor was any reference made to PEST settings.

*Response:*

Thanks for the comment. In our original version of manuscript, we used estimation mode for PEST, and considered equal weight for all four calibrated stations. There was no singular value decomposition (SVD) nor regularization used.

In our revised manuscript, we conduct the calibration using SVD-based regularization, we assign prior information for all the calibrated parameters, and we also consider different weights for the calibrated stations. We add the PEST setting in **Section 3.2 PEST files and settings**:

"This study presents calibration results from PEST using the SVD-based regularization in regularization mode to ensure numerical stability (Tonkin and Doherty, 2005). We focus on calibrating 22 parameters (see Table 1 and detail description in Sec. 3.3) using 96 observation points and 22 items of prior information for the calibrated parameters. In each item of prior information, a value equal to its default value provided by the WRF-Hydro v5.0 (or the log of its default value) is assigned for each adjustable parameter, assuming that default values are the preferred values. All prior information equations are assigned a weight of 1.0. We assigned five different regularization groups to the prior information: Manning's roughness coefficients specified by Strahler stream order in CHANPARM.TBL to one group; the parameters in HYDRO.TBL (Manning's roughness coefficients for overland flow as a function of vegetation types) to another group; and three global parameters for the Noah-MP (xslop1, refdk, and refkdt) in GENPARM.TBL to the remaining three groups. The 96 observation points are given different weights based on the inversed mean of their observed discharge during the studied period (see the detailed description in Sec. 3.3 and Sec. 4.1). For a detailed description of these settings see the PEST User Manual (Doherty, 2015)."

While I agree with the authors that use of inversion methods that can parallelize model runs and handle the estimation of many parameters employed by a complex model is a much-needed addition to the arsenal of surface water modelling, I think that many more advances could be made than the authors have made. In particular, there was no mention of the use of Tikhonov regularization to accommodate parameter nonuniqueness at the same time as it promulgates uniqueness through obtaining a set of parameters that "make sense" from an expert knowledge point of view. This, I think, is one of the strongest arguments for use of gradient-based, highly parameterized methods in regional surface or land use model calibration, that is the ability to not just accommodate nonuniqueness, but to turn the "wiggle room" engendered by nonuniqueness into formulation of an inverse problem that can actually make regionalization and transportability of parameters a reality.

*Response:*

Thanks for your comment, and we understand this is one of the major concerns for how PEST was used in our original manuscript to calibration a hydrological model and more broadly, environmental models. During the revision of this study, we conduct all the WRF-hydro calibration using SVD-based regularization. We also use prior information for the parameters that we calibrated, as you can see from our previous response about PEST settings. In **Section 5 Summary and discussion**, we also commented that, "In this study, we consider using the prior or regularization information only for the parameters that we calibrate. As is the case with solving inverse problems, prior information is added to improve the smoothness of the solutions. In order to build a more comprehensive calibration, an important aspect that can be considered is to enrich the prior with the available historical data. For example, in this particular case, one can use the historical observation data (e.g., April and May from the past few years) to enrich the prior information for the parameters. Hence, the regularization objective function in PEST will constitute not only the discrepancies between parameters and their "current estimates" but also the discrepancies between WRF-Hydro simulations and preferred values (which is the observed time series of historical discharge). Additionally, one can use the pilot points technique described by Doherty (2005) in conjunction with parameter estimation to add more flexibility to the calibration process. This will be potentially beneficial in improving the predictions"

In addition, to emphasize the importance of regularization for hydrological and environmental model calibration, we mentioned regularization in introduction: "PEST has four modes of operation. One of the modes is regularization mode, which supports the use of Tikhonov regularization and is found better for serving environmental models, because if implemented properly, it supports model predictions of minimum error variance, it is numerically stable and it embraces rather than eschews the heterogeneity of natural systems. Singular value decomposition (SVD) can be used as a regularization device to guarantee numerical stability of the calibration problem".

We also find that regularization mode does generate a better hydrograph shape compared to using estimation mode:

In **Section 4.1 WRF-Hydro calibration and validation**, we mentioned "Compared with the results of calibration using the estimation mode (no regularization) in PEST, the SVD-based regularization generates slightly better hydrograph shape with 24-hour later discharge peaks that are closer to the observations."

In **Section 4.3 Evaluation of spatial transferability of the calibrated parameters**, we mentioned "We also find that using the SVD-based regularization for the PEST calibration captures the timing of discharge peak better than using the estimation mode, which is one-day earlier than the observations reaching the discharge peak."

The authors use a simple objective function. This may be ok for some inverse problems. However as they point out, some of the smaller flows (in terms of location in space and location in a single flow time series) are not as well fitted as they could be. Perhaps weights should be a function of flow – and of location. Perhaps other important aspects of the flow time series should be made more visible to PEST through formulation of separate, targetted objective function components to ensure that these aspects of the time series are also well fit.

*Response:*

We understand this is another major concern about how PEST was setup for this study. We agree that, due to the fact that the calibrated station we chose are on different size in terms of water volume, to better handle the smaller river station (Station 1), considering different weight is fundamental, as also pointed out by Reviewer #2.

Therefore, during the revision of this study, we conduct all the PEST calibration of WRF-hydro considering a higher weight for Station 1 than for the other three stations. We add description about weight in the revised manuscript in **Section 3.2 PEST files and settings**, as you can find from previous response. We also add description in **Section 3.3 Calibration experiments:**

"We find that by using the same absolute weight for all four stations, the calibration helps three stations (Station 2, 3, and 4) with large water volumes to generate more reasonable results than do the default parameters; however, the results for Station 1, which has a relatively small volume of water, is not always better than the discharge that is modeled by using default parameters. Thus, we assign a weight of 15.0 for Station 1 versus a weight of 1.0 for the other three stations according to the inversed mean of observed discharge over these four stations in April 2013. The ratio of the weights between Station 1 and the other three stations stays similar even if the means are calculated based on different time periods."

We describe the results in **Section 4.1 WRF-Hydro calibration and validation** by comparing PEST calibration using equal weight and a higher weight for Station 1:

"Compared with the results calibrated by using equal weights for all the stations, by giving a higher weight to Station 1 the model bias over Station 1 is significantly reduced, with a higher NSE (0.87 with higher weight versus 0.14 with equal weight) and lower RMSE (48.1 versus 123.6)."

Here is a figure (Figure I) showing how the weight helps the result of Station 1. "Calib5" represent the results using higher weight for Station 1, while "Equal Weight" the results using equal weight for all the four stations.

[Figure]

The authors make a big deal out of their modifications to parallel PEST so that it is HPC-friendly. Actually, I think that the BEOPEST version of PEST has similar capabilities. The original version of BEOPEST used both MPI and TCP/IP for communication between master and slaves (now called manager and workers). Now only TCP/IP is used. One of the reasons that BEOPEST's capabilities exceed those of parallel PEST in the HPC environment (actually on any network) is that the manager does not need to write model input files and read model output files across the network. This makes run management must faster, more secure, and able to take place in a greater variety of network environments.

*Response:*

Thanks for the comment. Since our major objective is to build the bridge (or interface) between the parallel PEST and WRF-Hydro on the basis of HPCs, we expect the interface still working if one wants to use BEOPEST instead of parallel PEST to calibrated WRF-hydro, assuming there is also a version of BEOPEST in Linux environment (this is not clear to us by reading the manual). The only main changes for our script would be copying the template files and instruction file into each working directory. The command to execute BEOPEST is also different, which is *beopest* instead of *ppest* for parallel PEST. We add this paragraph in **Section 5 Summary and discussion**:

"While this study ported the parallel PEST to HPC system and linked it to WRF-Hydro, we note that BEOPEST is available in the PEST family. BEOPEST has the same functionality as parallel PEST but uses a different approach for communication between master and workers. Working with HPC-enabled BEOPEST may save total time cost since BEOPEST uses the Transmission Control Protocol (TCP) and the Internet Protocol (IP) instead of message files (reading input and writing output between master and works) for communication. We expect it to be relatively straightforward to use BEOPEST to calibrate WRF-hydro on HPCs since the interface remains similar, except one needs to copy the template and instruction files in addition to the global files (see Section 3.1) into each working folder."

In summary, I think that what the authors have done is good. However I also think that the potential for regional surface water model calibration and uncertainty analysis in a HPC environment still remains largely untapped. Some of this potential will be realized with use of singular value decomposition to ensure numerical stability when inverse problems are ill-posed, use of Tikhonov regularisation to ensure parameter sensibility and transportability under the same conditions, and more creative formulation of the objective function than the authors have done.

*Response:*

We thank Reviewer #1 again for all your valuable comments. Although optimizing the objective functions in PEST is beyond the scope of this study, we do have some thoughts for future studies. We add this in **Section 5 Summary and discussion**:

"In this study, we consider using the prior or regularization information only for the parameters that we calibrate. As is the case with solving inverse problems, prior information is added to improve the smoothness of the solutions. In order to build a more comprehensive calibration, an important aspect that can be considered is to enrich the prior with the available historical data. For example, in this particular case, one can use the historical observation data (e.g., April and May from the past few years) to enrich the prior information for the parameters. Hence, the regularization objective function in PEST will constitute not only the discrepancies between parameters and their "current estimates" but also the discrepancies between WRF-Hydro simulations and preferred values (which is the observed time series of historical discharge). Additionally, one can use the pilot points technique described by Doherty (2005) in conjunction with parameter estimation to add more flexibility to the calibration process. This will be potentially beneficial in improving the predictions."

**Reviewer #2**
The paper of Wang et al. deals with a potentially interesting implementation of the parallel version of the PEST software. PEST is a powerful and very useful tool for hydrologists, helping them during long and "exhausting" calibration sessions. Therefore, introducing the portability of parallel PEST to HPCs is good news and, specifically for the present paper, the main theme to highlight.

*Response:*

We are so glad that the reviewer finds this study helpful for hydrologist with the most tedious part of model development and application — calibration. We also wish to thank the reviewer for your great insight about the highlight of this study, which add important value to this manuscript.

Nevertheless, in my opinion the way the paper is structured mainly highlights, instead of the advantages of the novelty, the performances of the PEST calibration, which is something widely and well assessed by the hydrology research community. Almost all figures and tables deal with PEST results. Furthermore, the calibration procedure presented is questionable from different points of view (some of which are exposed later).

*Response:*

Thanks and we agree your comment. In our revised manuscript, we delete the section of 7-day calibration (figures and tables) considering it is unnecessary to support our main objective. We also delete or shorten the descriptions about WRF-Hydro and PEST, which readers can easily learn from the User Guides/Manuals. We thus focus on only two points: one is to show the operational feasibility, and the other is to explore the computational benefits of the HPC-enabled parallel PEST linked to WRF-Hydro. To demonstrate the first point, we calibrate WRF-Hydro for 3 days using SVD-based regularization method in PEST, and considering different weight for the four calibrated stations. For the second point, we design new experiments, using different computing resources for PEST workers and for WRF-hydro. Details can be found below in our response to your later comments.

The most interesting/innovative Section of the paper is Section 5.1, but the analysis of scale-up capabilities should be described with much more detail. Concerning the main outcomes of the paper highlighted in the summary, points from 2 to 5 are quite obvious (they deal with the recognized skills of the PEST software), while point 1 should be expanded: what does a factor of 30 "with respect to a serial calibration" exactly mean? In my opinion it's not a rigorous statement. What do the authors exactly mean with "serial"? Even though PEST calibration is serial, WRF-Hydro can run in a parallel fashion, and the speed of the calibration process would depend on the number of nodes used for the hydrological simulation. A possible idea is to provide hints about the trade-off between the number of nodes/CPUs used for running the parallel model (i.e., WRF-Hydro in this case) and the number of nodes/CPUs used for running PEST in a parallel fashion. I guess it depends somehow also on the dimensions of the domain (and no information is given here about the number of cells in which the basin is discretized, so the reader has no idea about the actual computational burden).

*Response:*

Thanks for your comment and suggestions. We re-write **Section 5 Summery and discussion** to only summarize the findings of the study, and then raise some key points that beyond the scope of this study but may inform future studies. In other words, we delete majority of the summary that appears in points 2 to 5 in our original manuscript.

In the revised manuscript we expand the discussion of scale-up capabilities of parallel PEST linked to WRF-hydro by designing more experiments using different computing resource for PEST workers and for WRF-hydro. We add **Section 4.2 computational benefits of parallel PEST on HPCs**. Some key notes about the experiments and our findings are quoted below:

"In this study we test the computational performance of HPC-enabled parallel PEST using different number of workers (6, 12, and 23) for the 22-parameter calibration. As shown in Table 3, we conducted five experiments: Test 1 uses 23 workers, Test 2 uses 12 workers, and Test 3 uses 6 workers. All three tests use two nodes for each worker to run WRF-Hydro in parallel."

"In order to test the trade-offs between the computing nodes used for running parallel WRF-Hydro and the workers used for running parallel PEST, Tests 4 and 5 use different number of nodes for each worker to run WRF-Hydro in parallel. Explicitly, Test 4 uses four nodes per worker, and Test 5 uses six nodes per worker. Both tests use six workers for running the parallel PEST."

"when we assign 12 (and 6) workers to parallel PEST, the time cost for calculating the Jacobian matrix is increased by a factor of 2 (and 4) compared with the time cost of using 23 workers. The time cost for the parameter upgrade stays similar for the three experiments because only one cycle of WRF-hydro simulation is conducted to test the Marquardt lambdas. As a result, the total time cost for Test 2 is ~1.5 times more than that for Test 1, and the total time cost for Test 3 is ~1.5 times more than that for Test 2 (Fig. 4b). By extrapolating the speedup curve shown in Fig. 4a and Fig. 4b, we expect the total time cost to be ~1516 minutes when using only one worker (or sequential mode), which is about 15 times slower compared with running the PEST in parallel mode using 23 workers."

"On the other hand, for the same case study and using the same number of nodes for running parallel WRF-Hydro, we can estimate the computing speedup by assuming an increase in the number of calibrated parameters to 50. This would be the case, for example, to evaluate model sensitiveness to the physics in Noah-MP or the spatial variabilities of certain parameters. We then expect to use 51 workers to achieve the best computing performance for parallel PEST. This would then be 28–30 times faster than running PEST using one worker (or in sequential mode). Similarly, if 100 parameters were used for the calibration for the same case study, a factor of up to 60 speedup in the calibration process would be achieved by running HPC-enabled parallel PEST."

"In addition, by increasing the number of nodes for each worker to conduct WRF-Hydro (Tests 3, 4, and 5), the time cost for the entire calibration process is significantly reduced (Figs. 4c and 4d). Specifically, the WRF-hydro scales up well when using four and six nodes compared with using two nodes per worker for running the WRF-Hydro. Both the time spent on calculating the Jacobian matrix and the time spent on testing the parameter upgrades are decreased by 49% and 67%, respectively, when using four and six nodes. Therefore, the total time spent is also decreased when using more nodes for each worker (see Table 3). Increasing the number of nodes to eight for each worker will most likely further decrease the time cost by 70–75% compared with using only two nodes per worker. Moreover, if one has a larger study area such as the entire contiguous United States, we expect the WRF-Hydro to have an even better scale-up capability (e.g., on dozens of nodes) than this study."

Here is a figure to show the computational benefit using parallel PEST to calibrate WRF-hydro.

[Figure]

Another important point, that should be better discussed, is the missed capability of the implemented version of PEST to deal with the calibration of spatially distributed parameters. This is important because it's reasonable to expect parallel PEST executions with WRF-Hydro for wide domains, and wide domains often need spatial differentiation of spatially distributed parameters, like, e.g., OVROUGHRTFAC, RETDEPRTFAC or other spatially distributed parameters available with WRF-Hydro v5.0.

*Response:*

We do acknowledge the importance of regionalization of parameter calibration, which definitely deserves future studies especially for large domains. We have been trying to add the interface on top of what we have now, to consider the spatial distributed files/parameters. For example, one can add regional OVROUGHRTFACs (e.g., their lower/upper bounds, and default values) in the PEST control file based on catchments/basins/regular regions etc. the potential challenge is that, the selection of the locations and sizes of catchment may introduce significant uncertainties to the calibration results. Thus it requires systematic and comprehensive investigation and understanding of the study area. We add a paragraph in **Section 5 Summary and discussion** about this:

"We only calibrate the parameters in lookup tables. Using a single value to represent a physics may work for a small domain but could be problematic for a large domain, especially we expect the HPC-enabled parallel PEST to execute with WRF-Hydro for large domains, which often need parameter regionalization. For example in WRF-Hydro v5, there are many spatially distributed parameters available such as the overland flow roughness scaling factor (OVROUGHRTFAC), the factor of maximum retention depth (RETDEPRTFAC), and the soil related parameters (when compiled with SPATIAL_SOIL=1). Calibrating these spatial parameters based on grid scale (e.g., catchments) rather than a single value will give the model more flexibility and thus can better fit the observations (Wagener and Wheater, 2006; Hundecha and Bardossy, 2004). In practice, for example, one can include regional OVROUGHRTFACs (e.g., their lower/upper bounds, and default values) in the PEST control file based on catchments. However, the selection of the locations and sizes of catchment may introduce significant uncertainties to the calibration results, which requires systematic and comprehensive investigation and understanding of the study area."

By the way, another limitation is that, at least as I understand, the calibration is available only against observed streamflow. Of course, this is the first option but not the unique one (one can decide to calibrate also against, e.g., soil moisture or latent heat flux data).

*Response:*

For the calibration exercise we did in this study, we use frxst_pts_out.txt as an instruction file which serves the format of output files of each working directory. In this file there is only discharge and water level available, so we calibrate the model using discharge data. It is feasible, however, to calibrate other variables as long as the observation data is available. For example, one can either find the closest point from the gridded dataset to the observation location and then compare that point to observations; or one can change the WRF-Hydro I/O code to output other variables in the frxst_pts_out.txt file, so they can still use the same interface we build here to calibrate other variables in addition to the discharge. We have added this regard in **Section 5 Summary and discussion**.

Finally, another important point is to (at least) discuss the problem of equifinality, which is incidentally (but not explicitly) dealt with in P11 L29 – P12 L5.

*Response:*

Thanks for your comment. It's actually interesting to think about this together with your other comment that one can also calibrate other variables rather than discharge. Since equifinality is an important source of model uncertainty, to reduce the model uncertainty, one may calibrate the model using multiple variables instead of one variables. This way the calibrate can constrain the model drift and may reduce the model uncertainty of prediction of certain variables (e.g. discharge and soil moisture). We add a paragraph in the **Section 5 Summary and discussion**:

"The optimal parameter set obtained from this study is from the 5th iteration of parallel PEST by testing five Marquardt lambdas. Testing different number of lambdas or calibrating different number of parameters may generate a different set of optimal parameters. These parameter sets can all make physical sense and be equally good for reproducing observed discharges. This problem is named equifinality (Beven and Freer, 2001; Savenije, 2001), which is an important source of model uncertainty. To reduce the model uncertainty through reducing the equifinality, hydrologists carry out additional modelling objective for model evaluation to find more useful parameter sets (Mo and Beven, 2004; Gallart et al., 2007). Alternatively, inspired by No. 3 discussed above, one can calibrate the WRF-hydro model based on more than one variables, such as discharge and soil moisture (or heat flux or water table depth) to reduce the number of optimal parameter sets, and thus reduce the model uncertainty of predictions for these variables."

Summarizing, though I acknowledge that the research presented is potentially interesting and innovative, I suggest to re-think the paper highlighting much more the computational benefits provided and reviewing the calibration performed in the case study.

*Response:*

We thank Reviewer #2 again for all your great insights. Our responses and revisions can be found above and below, as well as in the revised manuscript with tracked changes.

Following, a (not comprehensive) list of doubts regarding the calibration procedure and other minor comments and typos. I hope my comments can help improving the research.

Doubts about the calibration procedure:

Even though I acknowledge that authors decided to "focus less on extensively assessing the performance of the WRF-Hydro model", several aspects of the calibration procedure are very questionable.

1. no information about spin-up. This is extremely important, especially for such a short range calibration (only few days). The model should be run in advance (at least one month, I would say) in order to let several variables (e.g., moisture fields) have a realistic spatial distribution.

*Response:*

We did run the model for 3 months for spin-up. We apologize for not including it in the model description. Here is what we add in the revised manuscript:

"We start the WRF-Hydro simulation on Jan 1 2013 and run the model for more than 3 months to reach equilibrium. This 3-month period is considered as spin-up time and is excluded from model calibration and evaluation. We calibrate the river discharge calculated by the WRF-Hydro model from 00UTC April 9 to 00UTC April 12 2013, considering it is long enough to achieve our objective. We then evaluate the model performance against U.S. Geological Survey (USGS) observed river discharge from 00UTC April 12 to 00UTC 25, 2013."

2. the authors state that: April 8-11 moderate rain, April 12-14 no rain, April 15-18 rain, peak flow April 19. 3-day calibration is: April 9-11 (to be precise, April 12 at midnight), then validation is April 13-23 (April 12 is missed). 7-days calibration is April 9-15, validation is April 17-23. To me, it does not make too much sense that 4 more days are added when only the last one is rainy. It would be much better to calibrate the model with respect to a previous flood event, as it is usual. After all, observing graphs in figures 3 and 5 one after another just shows that increasing the number of days used for calibration improves the performances (but this is rather obvious), even though not yet enough.

*Response:*

As we mentioned earlier, in the revised manuscript we only focus on model calibration during April 9-11, and validation from April 12-24. We delete the 7-day calibration/validation results considering it is not necessary nor helpful to demonstrate our main objective. We add these sentences to emphasize this regard:

"The primary objective of this study is to build a bridge for linking the parallel PEST and WRF-hydro on the basis of HPC clusters and to explore the computational benefits of this bridge. We do not attempt to extensively assess each individual tool or address questions in each individual domain, such as optimizing the objective functions in PEST or calibrating WRF-Hydro for a long time period considering all the relevant parameters to achieve an optimal parameter set. The calibration period thus is limited to only three days, which we believe long enough to achieve our objective and to understand WRF-Hydro's sensitivity to the calibrated parameters."

We also add text in **4.1 WRF-Hydro calibration and validation** to explain the reason for the bias in hydrograph shape, such as the early peak and the faster decrease of river discharge:

"For Station 1, the WRF-Hydro almost captures the timing of the peak of discharge, although it still underestimates the water volume by ~25%. The reason is that this study uses a direct pass-through baseflow module, which does not account for slow discharge and long-term storage of the baseflow. Therefore, the largest contribution to river discharge is from precipitation, and groundwater does not contribute much discharge to the channels in a long-term view, as is also true for the other three large river stations. Different from Station 1, for the other three large river stations, the WRF-Hydro modeled discharge increases soon after the peak of precipitation and reaches a peak on April 21, 2013, which is much earlier than the observed peak of river discharge (near April 24). The reason is that the water contributions for these stations are from a larger river basin (Mississippi River) than we included in our current study area. Thus, when a heavy precipitation event occurs over the entire river basin, there will be a significant lag time (especially at the lower part of the basin) between the peak of precipitation amount and the peak of river discharge. For example, the precipitation over the upper part of Mississippi River Basin (MRB) has a peak amount on April 18–19, but the river discharge did not reach its peak until April 24. Because our studied area covers only half of the MRB, the modeled river discharge has a shorter delay period after the peak of precipitation than does the observed river discharge. Enlarging the study area to include the entire MRB may improve this situation. Alternatively, calibrating and validating local rivers that are included in the current study area may also reduce the bias in hydrograph shape compared to calibrating and validating large rivers. On the other hand, the WRF-Hydro simulated river discharge decreases soon after it reaches the peak and much earlier than the observed discharge. The reason is again that the direct pass-through baseflow employed by this study does not account for slow discharge and long-term storage of the baseflow. As a result, the contribution from the baseflow to the river discharge in model simulations does not stay as long as in real situations. In the observations, the river discharge decreases from the peak at a speed of ~500 m3/sec per day, while the modeled river discharge decreases from the peak at a speed of ~1667 m3/sec per day. Using exponential storage-discharge function for the baseflow may improve this situation."

3.  In order to deal with the observed streamflow in Section 1, it is fundamental to work with weights.

*Response:*

Thanks for your comment. This is also a major concern of Reviewer #1. For all the experiments we present in the revised manuscript, we consider a higher weight for Station 1. We add description about weight in the revised manuscript in **Section 3.2 PEST files and settings**, and in **3.3 Calibration experiments:**

"The 96 observation points are given different weights based on the inversed mean of their observed discharge during the studied period (see the detailed description in Sec. 3.3 and Sec. 4.1)."

"We find that by using the same absolute weight for all four stations, the calibration helps three stations (Station 2, 3, and 4) with large water volumes to generate more reasonable results than do the default parameters; however, the results for Station 1, which has a relatively small volume of water, is not always better than the discharge that is modeled by using default parameters. Thus, we assign a weight of 15.0 for

Station 1 versus a weight of 1.0 for the other three stations according to the inversed mean of observed discharge over these four stations in April 2013. The ratio of the weights between Station 1 and the other three stations stays similar even if the means are calculated based on different time periods."

We describe the results in **Section 4.1** by comparing PEST calibration using equal weight and a higher weight for Station 1:

"Compared with the results calibrated by using equal weights for all the stations, by giving a higher weight to Station 1 the model bias over Station 1 is significantly reduced, with a higher NSE (0.87 with higher weight versus 0.14 with equal weight) and lower RMSE (48.1 versus 123.6)."

A figure is shown above in the response to Reviewer #1.

Minor comments, grammar and typos

P6 LL6-17: not clear if in this case overland flow is switched on. It should.

*Response:*

Yes, it is. We change the sentence to emphasize this regard: "Overland flow, saturated subsurface flow, gridded channel routing, and a conceptual baseflow are active in this study"

"If overland flow is active as it is in this study, it passes water directly to the channel model."

P6 L19: probably "tools"

*Response: corrected*

P7 L18: GENPARM.TBL

*Response: fixed*

P8 LL11-12: master, not mater. The full stop is missing.

*Response: fixed*

P8 L30: As it is a common problem, it is usually solved 'simply' reallocating manually the stations. It's a pity to miss streamflow data for this reason

*Response:*

We regret we didn't do that, and it should be done in future applications. To clarify and to emphasize this regard, we change the sentence to:

"We calibrated WRF-Hydro using four USGS sites (referred to as Station 1, Station 2, Station 3, and Station 4 hereafter), as shown in Fig. 1. More USGS sites could be included if one manually reallocated the stations that were not properly assigned to the desired location on the channel network by the GIS tool."

P9 L24 and following: I suggest to explicitly declare also the meaning of the ovn parameter

*Response: added the vegetation type in Table 1 for each ovn\* parameter*

P12 L17: 50%, maybe

*Response: Apologize for the wrong number. It should be 66.7%*

Figs.2 and 3: April 12 is missing. It should be the first validation day, I guess.

*Response: added.*

Figs. 4 and 5: the same for April 16

*Response: this figure is deleted as it is for the 7-day calibration result*

Table 3: the note is incorrect, it refers to information about the 3-day calibration

*Response: fixed*

Section 4.3: this is a purely "hydrological" analysis that could be skipped, given the numerous limitations of the calibration procedure and the focus on the implementation of the PEST software

*Response:*

We still keep this section and the figure in the revised manuscript. One of the reasons is that, it is more obvious using these stations to show the benefit (generate better hydrograph shape) of SVD-based regularization compared to non-regularization, which is the method we emphasize that should be applied for calibrating hydrological and environmental models. The other reason is that, these rivers are relatively small and are local rivers that included in the current study area, therefore, (1) the lag time between precipitation peak and discharge peak is much shorter than those for Station 3, 4, 5, and the hydrograph shape is well captured by the optimal parameter set; (2) the slow-discharge effect from baseflow is also relatively small so the discharge decrease faster after the peak than that for Station 3, 4, 5. This is also well captured by the optimal parameter set. Overall, the WRF-Hydro calibration with current configuration actually did a good job to capture the hydrograph features for these stations.

P16 LL9-10: please check the sentence

*Response: this sentence is deleted, and changes are made accordingly through the entire manuscript.*

P16 L18: to investigate

*Response: corrected.*

[revised manuscript text omitted]

---

## Author Response (AR2)

Dear Editor Kurtz and Reviewer #2,

We would like to thank you again for the helpful comments and suggestions. In this version of revised manuscript, we did following changes to address the comments and suggestions:

1. We adjust the observation of streamflow for the model domain, and re-do the WRF-Hydro calibration for the same study area and same stations. Tables and figures are all updated accordingly.
2. Alternatively, we also calibrate WRF-Hydro over the same study area but using local river stations that are included in the study area. Tables and figures are presented in Supporting Information.
3. We present scalability analysis for the computational benefits of the HPC-enabled PEST coupled with WRF-hydro. A new figure is added. Table 3 is extended to include more scenarios.
4. Detailed comments are added in the published code to advise the use of developed scripts on other management software or job schedulers.

Please find our one-on-one response below to both editor's and reviewer's comment. A complete list of the changes made for the revised manuscript can be found in the "track changes" version of the manuscript. A clean version of the revised manuscript is also attached at the end.

Sincerely, Jiali
Wang
jialiwang@anl.gov

From Editor Kurtz:

(1) A more detailed analysis of the scaling behaviour and the optimal configuration with respect to workers and nodes per WRF-Hydro instance should be performed.

From Reviewer #2:

Concerning the analysis of computational benefits of parallel PEST on HPCs (new Section 4.2), the analysis performed is interesting, but to me it should be still improved, because does not yet respond clearly to the main question: "How can I use the nodes I have available in the most efficient way?" If I understand this section, based on Table 3 the Test 5 is better than the Test 1 because: Test 1) 23 workers x 2 nodes = 46 nodes and 103 min; Test 5) 6 workers x 6 nodes = 36 nodes and only 86 min. On the one hand, this result should be highlighted. But, on the other hand, the same result cannot be separated from considering the extent of the WRF-Hydro domain (from this point of view, I note that information about computational domain is still missing). Therefore, in general I would say that: first, a scalability analysis of WRF-Hydro over the specific computational domain should be performed; then, this scalability analysis should be used as a preliminary, but essential piece of information to provide comprehensive indications about the "optimal" configuration of the system for the analysed case study, given the threshold of the computational resources available.

**Response:** Thank you both for the great suggestion, which really adds value on our existing analysis about the computational benefits of the HPC-enabled PEST coupled with WRF-Hydro. The domain size had been mentioned a couple of times in Section 2.1 and 2.2. We emphasize it again in this version:

"The domain size is ~495,000 km2 (747 km from west to east; and 657 km from south to north)."

"…hydrological routing is performed at a grid resolution of 200 m, with 3285 south-north × 3735 west-east grid cells"

We add following discussion in Section 4.2 (Computational benefits of parallel PEST on HPCs) and a new figure (Figure 5) in the revised manuscript to demonstrate the scalability analysis. We also did one more experiment using 6 workers and 8 nodes per worker. Based on all the tests we did (Tests 1-6), we provide more scenarios about their time and computing cost when using different number of workers and nodes per worker by extrapolating the existing numbers (Table 3).

"While these numbers in Table 3 and Figure 4 are helpful to demonstrate the scale-up capability of each component (PEST and WRF-Hydro), they do not answer questions such as, if one has certain number of nodes, how many workers and how many nodes per worker should be used to achieve the highest efficiency of the WRF-Hydro calibration using HPC-enabled PEST? On the other hand, one may have unlimited computational resource, but would like to complete the calibration in a short time period. We present scalability analysis below to answer these questions. First, we generate more scenarios using different number of workers and nodes per worker by extrapolating the existing time and computing costs based on the experiments that are already conducted. These scenarios use 23 or 12 workers, and 4, 6, or 8 nodes per worker, respectively. Since we have conducted simulations using the same number of nodes per worker, the cost for these scenarios are easily predicted.

As shown in Figure 5, compare with Test 3 (which requires the least computing resource —12 nodes in total), having more workers (with the same number of nodes for each worker, e.g., Tests 1 and 2), takes more time than the ideal curve. The ideal curve assumes a linear speedup based on the time cost of Test 3. However, using the same number of workers and increasing the number of nodes for each worker (e.g., Tests 4, 5, and 6) can achieve the ideal speedup. Even when using 12 workers, increasing the number of nodes for each worker can still achieve a speedup close to the ideal curve. Using 23 workers will not achieve the ideal speedup. Therefore, if one only has a certain number of nodes available, we recommend to use relatively small number of workers but large number of nodes for each worker. For example, if one has 48 nodes, then there are three options can be considered: using 23 workers and 2 nodes per worker; 12 workers and 4 nodes per worker, and 6 workers and 8 nodes per worker. Other partition (16x3; or 8x6) between numbers of workers and nodes per worker are not as efficient as above. These three options will cost 103, 72 and 60 min, respectively, to finish one iteration. Thus, using 6 workers and 8 nodes per worker is the most efficient way to consume the limited computing resource. On the other hand, if one would like to conduct the calibration in a short time period without any limits for the computing resource, then using 23 workers and 8 nodes (perhaps even more nodes depending on the size of the model domain and the scale up capability of WRF-Hydro), will finish one iteration in ~24 min."

[Figure]

Figure 5. Total time cost and total computing resource needed for each test and extrapolated scenario, which uses different number of workers and different number of nodes per worker. The dash line is an ideal curve, which assumes a liner decrease in terms of time cost when more computing resource is used, built on Test 3. The circles are real cost for time and computing resource by each test and extrapolated. The red text and solid circles indicate those specific tests meet the ideal expectation of speedup.

From Editor Kurtz:

(2) The adopted calibration procedure as well as the description of the catchment models needs to be improved. Please see the comments from reviewer #2 for more details.

From Reviewer #2:

Concerning my doubts about the calibration procedure, the explanation provided: "The reason is that the water contributions for these stations are from a larger river basin (Mississippi River) than we included in our current study area" confirms that the calibration performed over stations from 2 to 4 does not make too much sense. Of course enlarging the study area to include the MRB will improve results, and this must be done if the authors still want to consider stations from 2 to 4. Another option, maybe more feasible for the authors, is to skip stations from 2 to 4 and consider stations from 5 to 8 not only for "transferability", but also for validation (but this must be done in a persuasive way). However, since from the first version of the manuscript it was not clear that the study area "covers only half of the MRB", and in the second version the authors only hint at it, it is very useful, for the sake of clarity, that the characteristics of all the catchments upstream the considered stations are clearly showed in the paper, both in a Table (e.g., indicating extent of the catchment and other main features) and in Figure 1, highlighting the borders of the catchments.

**Response:** Thanks for your comment, which motivates us to think about addressing this issue in two different approaches. First approach: as we presented in the revised manuscript, we adjusted the observation of streamflow for Stations 2, 3, and 4 by excluding inflows from catchments that are not covered by the study domain; Second approach: we also mentioned it in the revised manuscript, but provided results in Supporting Information. We calibrate the model against local river stations which has smaller drainage area and are included in the study area.

Below are the description that we include in the revised manuscript. This is for the first approach. We also edited the Figure 1 to demonstrate the idea, and highlight the borders of the catchments that we work with.

"As shown by the lower left index map in Figure 1, the study area (the red box) only covers the lower part of Upper Mississippi River Basin (UMRB) and a portion of Missouri River Basin (MORB). In order to prepare observation datasets of streamflow contributed only from the drainage area within the model domain, we identified inflows entering the model domain at three different sites, namely, sites 05411500, 06807000, and 06887500, as indicated by the black solid triangles in the index map of Figure 1. The outflows of combined UMRB and MORB can be found at the three outlets, namely, sites 07010000, 07020500, and 07022000 (named Stations 2, 3, and 4, respectively, as shown by black solid circles in Figure 1). These outlets are located sequentially at the main Mississippi River after confluence of Mississippi River and Missouri River. Thus, the observed streamflow contributed by drainage area within the model domain can be calculated by subtracting the sum of the discharge at the three sites (black triangles; recognized as inflow) from the discharge at each of the three outlet sites (black circles; recognized as outflow). The final derived observations of streamflow (or adjusted streamflow observation data) from the drainage area within this model domain are prepared for model calibration and validation. To prove this concept, we validated the consistency of the sum of observed drainage areas at inflow sites plus modeled drainage area with the overall drainage area at the outlet. The drainage area (UMRB and MORB) at outlet site 07010000 is 1.8E+12 m2. The sum of drainage areas at three inflow sites is about 1.4E+12 m2 (2.0E+11, 1.1E+12, and 1.4E+11 m2 for site 05411500, 06807000, and 06887500, respectively) and the modeled drainage area is 0.36E+12 m2; the total area is 1.76+12 m2. This indicates that the flows from sum of three inflow sites and modeled result represent 98% of drainage area at the outflow site 07010000. Therefore, the adjusted streamflow observation data are qualified for model calibration."

For the second approach:

"Alternatively, instead of calibrating the stations that have large drainage area and water coming from outside of the current model domain, we have also tested calibrating small flows at local stations that have relatively small drainage area covered by the current study area. This requires to generate a new high-resolution GIS data file to distribute the stations of interest. We first run the WRF-Hydro model for 6 month using default parameters to spin up the model, and then we calibrate the model based on observations of these local stations. Results including figures and tables are shown in Supporting Information. The calibration results are improved compared to the results that use default parameters, although further improvements are still needed. This again may be because the parameter range are not wide enough to consider the possible values of parameters that work for these specific areas represented at local stations, as we see many optimal parameters hit the bound of the parameter range. More tests to figure out a better set of parameters are needed for future investigation, which is beyond the scope of this study, since our goal is to present the feasibility of HPC enabled PEST."

[Figure]

Figure 1:  Eight USGS sites over the study area (750 km x 660 km). The four circles are sites that are used for calibrations; the four crosses are sites that are used for transferability assessment. USGS site numbers corresponding to the site index used in this study are: Station 1: 05465500; Station 2: 07010000; Station 3: 07020500; Station 4: 07022000; Station 5: 05465700; Station 6: 05474000; Station 7: 05558300; Station 8: 05568500.  The three inflow stations indicated by the black solid triangles are 06807000, 06887500, and 05389500.

**Table S3: Statistics of model performance using optimum and default (in parentheses) parameters for local stations during the calibration period.**

| Statistics | 0556855 | 05474000 | 05465500 | 05558300 |
|---|---|---|---|---|
| | | Calibration | | |
| NSE | 0.30 (-0.46) | -8.5 (-46.6) | 0.45 (-1.28) | -7.11 (-16.6) |
| RMSE | 431.88 (624.95) | 235.25 (526.44) | 351.18 (716.0) | 1579.9 (2329.9) |
| PCC | 0.96 (0.30) | 0.54 (-0.86) | 0.78 (0.33) | 0.70 (-0.23) |

[Figure]

**Figure S2: Observed and modeled discharge (m³/sec) using default and calibrated parameters during a 3-day calibration period (April 19–21, 2013) over four local stations. Station numbers are indicated on top of each panel.**

From Editor Kurtz:

(3) I would also like to encourage you to provide a bit more details on how the interface between WRF-Hydro and PEST works on a technical level (i.e., add a bit more description on what happens in the developed script) and maybe add a short discussion on what would need to be done to use the interface on other systems than the ones described in the paper. This would help potential users to adopt the described interface to their own system.

**Response:** Thank you for the suggestion. The script we developed to bridge the parallel PEST to WRF-hydro on HPC is fairly straightforward and easy to use. While we had described what the script does, in this version, we emphasize that, if it is going to be used on a different management software using a different job scheduler, the only two things that users need to figure out are: (1) how to find and identify available nodes, and (2) how to submit a regular job on that specific server. Although the experiments are presented are conducted on SLURM manager and job scheduler, we have also provided scripts that have been tested and work well on Cobalt manager and job scheduler, which provide the differences in scripts that operate on different HPCs. We have also added detailed comments in the published code (http://doi.org/10.5281/zenodo.3247116) which advise users to make the changes if needed. We are also open to be contacted if users need help from our side. We look forward to working with WRF-Hydro users to make this tool feasible on different HPCs and for different user cases.

We would like to thank the reviewer and the editor again for your suggestions and comments, which tremendously improve this study.

[revised manuscript text omitted]